# Beyond Static Forecasts: A Dynamic Stress Gradient Framework for High-Resolution Aftershock Prediction and Mitigation

Boi-Yee Liao<sup>1</sup>

- <sup>1</sup> Department of Engineering Management, International College, Krirk University, Bangkok 10220, Thailand Department *Correspondence to*: Boi-Yee, Liao (y5708211@ms18.hinet.net)
- Abstract. Accurate forecasting of aftershock distributions is vital for effective post-earthquake emergency response, early warning systems, and long-term seismic hazard mitigation. This study introduces a novel nonlinear, multiscale framework for modeling the evolution of Coulomb stress following a major earthquake. The proposed approach integrates rate-and-state friction laws, a KPP-type reaction–diffusion equation, and the Banach fixed-point theorem to simulate the dynamic redistribution of stress in space and time. Central to the model are two time-dependent parameters— $\alpha(t)$ , which governs the decay of stress memory consistent with Omori's law, and  $\beta(t)$ , which modulates the nonlinear diffusion and reaction dynamics. Applied to the 2018 Hualien earthquake in Taiwan, the framework resolves stress changes and their gradients at depths ranging from 6 to 25 km. Results indicate that stress gradients are more predictive of aftershock occurrences within the first 50 days and at depths shallower than 12 km, while stress changes play a dominant role at greater depths and later times. Validation using AUC and Molchan error metrics demonstrates the model's strong spatial forecasting capability. The framework's adaptive convergence and modular structure support real-time seismic hazard assessment and integration into PSHA workflows, offering a promising tool for aftershock modelling and disaster resilience planning.

## 1 Introduction

20

Developing a precise stress evolution model that can rapidly predict the timing and location of aftershocks following a massive earthquake is of immense value. Such a model would enhance disaster preparedness, optimize resource allocation, and mitigate the devastating effects of aftershocks, ultimately safeguarding lives and reducing the risk of further losses caused by aftershock damage. The occurrence of aftershocks is intricately tied to stress changes induced by the mainshock, and the evolution of post-seismic stress over time plays a pivotal role in predicting aftershocks (Devires et al., 2018; Aden-Antoniow et al., 2022). Furthermore, seismic stress evolution is fundamental to advancing our understanding of earthquake mechanics and the associated risks. The complex interplay of processes governing stress accumulation and release in the Earth's crust underscores the need for more comprehensive modeling approaches. A modeling framework that can simulate post-seismic stress redistribution and identify potential earthquake-triggering zones is essential for improving seismic risk assessment and management. However, despite much progress, a comprehensive theory explaining how post-earthquake stress evolves and leads to aftershocks remains elusive.

30

In the evaluation of aftershock locations, the primary focus is on the application of Coulomb stress change. The Coulomb stress model has been a foundational tool for understanding fault interactions and stress transfer during seismic events (King et al., 1994). By quantifying shear and normal stress changes, it evaluates aftershock potential and fault stability under diverse tectonic conditions (Harris, 1998). Early foundational work by Mignan and King (2007) formulated accelerating moment release based on stress accumulation and transfer models, establishing a framework for exploring stress interactions. Subsequent studies expanded the application of the Coulomb model to various geological contexts, such as stress evolution in subduction zones (Hsu et al., 2006) and fault systems like the Dead Sea Fault (Heidbach and Ben-Avraham, 2007) and Tien Shan (Pang, 2008). Specific earthquake events further illustrate the model's utility. For example, research on the 1999 Chi-Chi earthquake in Taiwan revealed how stress redistribution influences afterslip, relaxation, and deviations from Omori decay (Chan and Stein, 2009). Analyses of the 2008 Wenchuan earthquake demonstrated cascading stress effects on subsequent seismic events (Zhang et al., 2013), while viscoelastic relaxation studies enhanced understanding of stress redistribution in Sichuan (Xie et al., 2022). Similarly, the Coulomb stress models of earthquake sequences have underscored how historical stress changes can influence fault behavior and trigger seismic activity, respectively (Nathan and Walter, 2020; Tang et al., 2023; Ueda and Kato, 2023). Long-term investigations, such as those on the East Kunlun Fault Zone (Shan et al., 2015) and the Sulaiman Lobe (Ali et al., 2017), highlighted the model's potential for assessing regional seismic hazards. While the Coulomb stress model has significantly advanced our understanding of stress transfer and fault interactions, its typical applications often focus on isolated events or discrete timeframes. This limitation restricts its ability to capture the continuous evolution of stress fields, which is essential for modeling dynamic aftershock sequences and improving seismic hazard assessments.

Secondly, combining the Rate-and-State (R-S) physical model to explore stress transfer near faults and understand the mechanisms of aftershock generation represents an advanced development. The R-S friction model, first introduced by Dieterich (1994), has been instrumental in describing the relationship between stress and friction on faults. Grounded in Coulomb stress and friction constitutive laws, the model provides a robust framework for investigating earthquake nucleation and fault mechanics. Early work by Ampuero and Rubin (2008) explored aging and slip laws within the R-S framework, offering critical insights into the processes governing earthquake initiation. Building on this foundation, Barbot et al. (2012) advanced the model by integrating short-term earthquake dynamics with long-term geological processes, enhancing its capacity to simulate seismic cycles and their broader implications. Recent applications highlight the model's adaptability to diverse tectonic settings. Javed et al. (2016) integrated Coulomb failure and R-S models to replicate the Omori-Utsu relation for aftershock decay, shedding light on stress shadowing effects and the temporal evolution of seismicity. Pranger et al. (2022) further expanded the model by combining R-S friction with transient viscous flow principles, resulting in a more comprehensive representation of fault dynamics. Su et al. (2024) applied R-S friction principles to estimate earthquake probabilities along the Liupanshan Fault, demonstrating its potential for probabilistic seismic hazard assessments. Despite these advances, the R-S model faces inherent challenges in addressing spatiotemporal complexities of stress evolution. High friction parameters can lead to numerical divergence, limiting the model's applicability in






heterogeneous tectonic settings. Furthermore, localized implementations, while computationally efficient, often lack mechanisms for stress diffusion, restricting their ability to capture interactions across larger fault systems. Overcoming these limitations requires the integration of additional diffusion processes and stability mechanisms to better address the interplay between localized and distributed stress evolution. Such advancements are essential for improving the model's predictive power and its applicability in seismic hazard assessments.

Finally, the distribution of aftershocks provides other valuable insights into seismic stress dynamics, with the widely recognized Omori Law describing the temporal decay of aftershock sequences (Utsu, 1961). While empirical laws offer foundational understanding, theoretical models have advanced this knowledge by incorporating reaction-diffusion frameworks, such as the Kolmogorov-Petrovsky-Piskunov (KPP) equation. Guglielmi et al. (2021) demonstrated the utility of the KPP equation for understanding stress propagation in seismic contexts. Regenauer-Lieb et al. (2021) further highlighted its application through cross-diffusion-driven wave processes, transforming the KPP framework into a dynamic wave-driven stress diffusion model that aligns its diffusion-reaction characteristics with the evolving stress field more effectively. Recent studies have reinforced the promise of the KPP framework for capturing aftershock evolution. For instance, Zavyalov et al. (2024) applied the KPP equation to model the spatial and temporal dynamics of aftershock distributions following mainshock events. Complementing these approaches, nonlinear equations such as the Fisher equation have been employed to model wave propagation and spatial diffusion in geological settings, further enhancing the understanding of stress distribution (Alqahtani et al., 2024).

However, despite its effectiveness in modeling stress diffusion, the KPP equation has notable limitations. It does not directly incorporate frictional behavior and is prone to numerical instability in heterogeneous tectonic environments, which restricts its broader applicability. To overcome these challenges, Banach's Fixed-Point Theorem (Granas and Dugundji, 2003) offers a robust mathematical framework for ensuring numerical stability. By conceptualizing the stress release process as an iterative contraction mapping, the theorem guarantees convergence to a stable state after a destructive earthquake. The integration of Banach's Fixed-Point Theorem with the KPP model marks a significant advancement over conventional stress modeling approaches. By resolving issues of numerical divergence and improving stability in nonlinear stress propagation scenarios, this combined framework provides a more comprehensive and reliable method for capturing the intricate dynamics of aftershock sequences and stress evolution.

Building on this foundation, this study introduces a novel integration of the Rate-and-State (R-S) friction model, the KPP equation, and Banach's Fixed-Point Theorem. The KPP equation effectively models spatial stress diffusion, the R-S friction model captures nonlinear stress accumulation and slip behaviors along rocks, and Banach's theorem ensures numerical convergence by addressing challenges in heterogeneous tectonic environments. This integrated framework overcomes critical challenges in simulating stress redistribution across depths and time scales, offering a robust approach for aftershock forecasting. This comprehensive framework not only advances the understanding of fault mechanics and stress evolution but also bridges theoretical advancements with practical applications in seismic hazard mitigation. By effectively modeling post-earthquake stress evolution and aftershock distributions, this approach facilitates precise identification of







high-risk areas and temporal patterns of seismic activity. Such insights enable the strategic deployment of emergency response teams, the optimized allocation of relief resources, and the prioritization of inspections and reinforcements for vulnerable infrastructure.

Moreover, incorporating this model into disaster preparedness frameworks enhances risk communication and raises community awareness, empowering residents to take proactive measures to safeguard their lives and properties. Over the long term, this methodology can guide the design and retrofitting of seismic-resistant infrastructure, strengthening communities' ability to withstand future earthquakes. By translating predictive insights into actionable strategies, this research bridges the gap between theoretical advancements and practical disaster resilience, promoting safer, more sustainable urban environments.

# 2 Methodology

#### 2.1 The iterative process of Coulomb's stress changes

This study integrates diffusion and reaction dynamics into a unified stress evolution framework. The methodology ensures numerical stability and convergence, enabling robust simulations of stress redistribution after earthquakes. When deriving a seismic source slip model, the initial spatial distribution of Coulomb stress changes,  $\sigma(x,t=0)$ , is calculated using the Okada method (1985) for specified depths and initial conditions:

$$\Delta CFF = \sigma(x,t=0) = \Delta \tau + \mu^{'}(\Delta \sigma_{normal}) \ \ (1)$$

where  $\Delta\tau$  is the shear stress change,  $\mu'$  is the effective friction coefficient which is generally set to 0.4 (King, et al., 1994),  $\Delta\sigma_{normal}$  is the normal stress change. In this research, the coseismic slip model used in the Coulomb stress change calculation was derived from Liao et al. (2024), which employed GPS data and GA inversion methods. The corresponding aftershock catalog was obtained from the Central Weather Bureau (CWB) of Taiwan, with hypocentral locations refined using double-difference relocation, and is publicly accessible at https://www.cwa.gov.tw/V8/E/E/index.html.

To model post-earthquake stress evolution, we adopt a reaction—diffusion framework inspired by the Kolmogorov—Petrovsky—Piskunov (KPP) equation (Zavyalov et al., 2022), which captures the interplay between stress propagation and local reaction processes:

$$\partial \sigma(x,t)/\partial t = D\nabla^2 \sigma(x,t) + f(\sigma(x,t),\theta(x,t))$$
 (2)

where D represents the diffusion coefficient, and the term  $\nabla^2 \sigma(x,t)$  denotes the Laplacian of the stress field, which describes stress propagation. The stress reaction term  $f(\sigma(x,t),\theta(x,t))$  is a nonlinear reaction term that reflect Rate-and-State (R-S) friction behavior as well as stress saturation at high-stress levels, accounting for both stress buildup and frictional weakening. To reflect the influence of fault healing and time-dependent frictional behavior, the reaction term includes a logarithmic dependence on the state variable  $\theta$ , inspired by Dieterich (1994) and Ampuero & Rubin (2008). It is presented as

$$f(\sigma, \theta) = A\sigma \left(1 - \frac{\sigma}{\sigma_{max}}\right) + Blog\left(\frac{\theta}{\theta_0}\right)$$
 (3)





The first term models stress accumulation and saturation, ensuring the reaction vanishes at both zero and maximum stress  $(\sigma=0 \text{ and } \sigma=\sigma_{max})$ , analogous to logistic growth. The second term captures the aging effect which means the longer the contact time (higher  $\theta$ ), the stronger the frictional resistance. Parameters A and B govern the magnitude of these effects and are allowed to vary with depth are described as

$$A = A_0 \exp(-\operatorname{depth} / L_a)$$
 (4)

$$B = B_0 \exp(-\operatorname{depth}/L_h)$$
 (5)

where L<sub>a</sub> and L<sub>b</sub> are characteristic depths that influence the variation of friction parameters with depth, accounting for the impact of rock properties at different depths on stress propagation. According to rate-and-state friction law (Dieterich, 1994), the differential of state variable θ is used to calculate slip rate V by the following formula

$$d\theta/dt = 1 - (V\theta)/L$$
 (6)

where L is the characteristic slip distance. The slip rate V depends exponentially on both stress and the state variable. V can be expressed as

$$V = V_0 \exp\left(\frac{A \cdot \sigma(x,t) + B \cdot \log\left(\frac{\theta}{\theta 0}\right)}{\sigma_{\text{max}}}\right) (7)$$

where  $V_0$  and  $\theta_0$  are constants initial values for the slip rate and the state variable, respectively. This formulation ensures that stress evolution is bounded, physically grounded, and sensitive to fault aging and local stress levels. The combination of Eqs. (2)–(7) constitutes a depth-aware, multiscale framework that couples nonlinear stress accumulation with memory-dependent frictional responses, enabling realistic modeling of aftershock generation mechanisms. To ensure physical consistency, the nonlinear reaction term in the stress evolution model is derived from rate-and-state (R–S) friction theory. It combines two key mechanisms: a stress saturation effect, which limits stress accumulation near rupture thresholds, and a logarithmic dependence on the state variable, representing frictional healing over time. These features reflect experimental observations that fault slip is governed both by current stress and by the time-dependent maturity of contact surfaces. The model also incorporates depth-dependent frictional parameters, decreasing with depth to account for variations in rock strength and healing efficiency. This formulation links the evolving stress field with realistic slip behavior, where aftershocks preferentially occur in zones of elevated stress and aged frictional contacts. By integrating R–S friction into a reaction—diffusion framework, the model captures the interplay between spatial stress propagation and temporal healing. This approach offers a more dynamic and physically grounded representation of postseismic processes than traditional static models, improving insight into aftershock triggering and stress relaxation.

To ensure both numerical convergence and physical realism in long-term stress evolution, we recast the reaction-diffusion equation into a contraction mapping form and apply the Banach Fixed-Point Theorem. The iterative stress update is expressed as

$$\sigma_{n+1}(x,t) = T(\sigma_n(x,t)) = \alpha \cdot \sigma_n(x,t) + \beta \cdot (D \cdot \nabla^2 \sigma_n(x,t) + f(\sigma_n(x,t),\theta_n(x,t)))$$
(8)

where  $\alpha$  is assumed to satisfy the condition  $0



stress saturation effects. For guaranteeing convergence, the mapping T must satisfy the contraction condition of Banach Fixed-Point Theorem. It is necessary to find a constant  $k \in (0,1)$  such that, for any two stress states  $\sigma$  and  $\sigma'$  of a stress field satisfying the condition

$$\parallel T(\sigma(x,t)) - T(\sigma^{'}(x,t)) \parallel \leq k \parallel \sigma(x,t) - \sigma^{'}(x,t) \parallel (9)$$

This leads to the constraint on  $\beta$  having the following relation

$$0 



Physically,  $\alpha$  captures the persistence of prior stress conditions, analogous to frictional healing effects in fault mechanics. A higher  $\alpha$  indicates greater retention of past stress, while a lower  $\alpha$  allows rapid adaptation. Conversely,  $\beta$  determines the strength of each update, acting as an effective step size for stress redistribution and state evolution. From a numerical perspective,  $\alpha$  acts as a damping factor, preventing abrupt transitions by controlling how much of the prior stress state is retained. Parameter  $\beta$  functions similarly to a relaxation parameter, scaling the response to local perturbations and ensuring convergence under Banach's condition. To clarify their dual roles, we provide Table 2 as below.

Table 2. The physical meanings and numerical roles of the parameters  $\alpha$  and  $\beta$ 

| Parameter | Physical Interpretation                 | Numerical Role                                    |
|-----------|-----------------------------------------|---------------------------------------------------|
| α         | Memory of historical stress             | Dampens prior stress influence in iteration       |
| β         | Strength of reaction-diffusion response | Scales updates; governs convergence and stability |

In the subsequent numerical simulations, we demonstrate how variations in  $\alpha$  and  $\beta$  influence convergence rate and stress field evolution. A balanced choice ensures both stability and efficiency while reflecting the natural damping behavior of crustal stress systems.

# 2.2 Numerical Stability and Parameter Constraints

To ensure numerical convergence and model reliability, we adopt spatial grid resolutions of  $\Delta x = 1320$  m and  $\Delta y = 2600$  m, yielding an effective length scale of  $\Delta x_{eff}$ =2900. The stress diffusion coefficient is set as D=10<sup>-2</sup> m<sup>2</sup>/s, consistent with fluid-assisted or elastic stress propagation in the crust. The corresponding term DC~1/ $\Delta x_{eff}^2 \times 10^{-2}$  is approximately to  $10^{-8}$ , quantifying the smoothing effect of diffusion on the evolving stress field. To satisfy the contraction condition  $\beta 



This duration aligns with the early aftershock window targeted in this study. The selected spatial resolutions were calibrated to balance model accuracy and computational feasibility. A sensitivity analysis with  $\pm 20\%$  variation in  $\Delta x$  and  $\Delta y$  confirmed that our configuration maintains CFL compliance ( $\beta$  upper bound = 95.3 for  $\pm 20\%$  and 42.4 for  $\pm 20\%$ ), validating the stability of our  $\beta = 50$  setting. While slight variations in grid resolution may impact Coulomb stress gradients and aftershock localization, our preliminary assessments show negligible distortion, supporting the robustness of our chosen setup.

## 2.3 Critical Iteration Number for Diffusion Dominance

I define the critical iteration number n<sub>critical</sub> as the point at which stress diffusion overtakes local stress reaction in driving the evolution of the stress field. This threshold is analytically derived as:

$$n_{critical} = \frac{\ln\left(\frac{A_0 \Delta \sigma_0}{D_0 \sigma_0}\right)}{-\ln(\alpha)} (15)$$

Here,  $A_0$  is the friction parameter,  $D_0$  is the stress diffusivity,  $\sigma_0$  denotes a characteristic stress amplitude, and  $\Delta\sigma_0$  is the initial stress contrast between adjacent grid points. Physically, this quantity marks the transition from a reaction-dominated regime (early stress accumulation) to a diffusion-dominated regime (stress homogenization). Assuming  $D_0 \approx A_0$  for simplicity, Eq. (15) shows that large stress gradients (i.e., high  $\Delta\sigma_0/\sigma_0$ ) lead to higher  $n_{critical}$ , particularly at shallow depths where heterogeneity is pronounced. In deeper layers, reduced gradients yield smaller  $n_{critical}$ , allowing stress diffusion to dominate earlier. Thus, this parameter serves as a diagnostic for identifying the timing of dominant physical processes in postseismic stress evolution.



## 220 3 Results and Discussions

#### 3.1 Detections of the proposed model

I applied the proposed iterative stress evolution model to the Mw 6.4 Hualien earthquake that occurred on February 6, 2018, at a depth of 10 km in northeastern Taiwan (Figure 1).

Figure 1: The epicenter of the moderate earthquake that occurred on 02/06/2018 is marked by a red star, while aftershocks ranging from magnitudes 2 to 5 within a three-month period are indicated by vellow circles.

The Hualien Mw 6.4 mainshock occurred in the complex convergence boundary of the Philippine Sea Plate and Eurasian Plate. According to the results of Huang et al. (2019), the Hualien earthquake was likely caused by three faults, with the greatest impact being on the west-dipping fault. This eventually triggered the shallower Meinong Fault, leading to surface rupture. Therefore, it is essentially an interplate event, which is consistent with its vigorous aftershock sequence. In the first 12 days alone, over 2,100 aftershocks were recorded in the Hualien sequence (Hao et al., 2018), reflecting high aftershock productivity characteristic of plate-boundary earthquakes. The earthquake resulted in 17 fatalities, over 300 injuries, and the collapse of four significant buildings (Nieh et al., 2020). The 2018 Hualien earthquake's aftershocks were mostly confined to mid-crustal depths (~5–15 km) with very few events in the uppermost 





lower crust), pore-fluid pressure build-up can induce slip instability on faults that would otherwise creep stably (Wen et al., 2019). Thus, at about 15–25 km depth, the rock chemistry and rheology (e.g. dehydration of minerals, crystal plasticity) can limit aftershock productivity unless high fluid pressures locally enable brittle failure. In the Hualien sequence, this is reflected by the paucity of aftershocks below ~15 km, suggesting that deeper rocks deform more aseismically. Jian et al. (2018) also showed that there were at least 16 earthquakes with a magnitude greater than 4.5, including one earthquake with a magnitude of 6.1, distributed in the depth range of 3 to 15 kilometers. In addition, Shallow aftershocks may decay more rapidly in number due to rapid stress relaxation in brittle, fractured rock, whereas at intermediate depths the decay could be influenced by fluid diffusion and healing processes in the fault zone. This deviation is likely due to different stress persistence in stronger rocks and possible ongoing creep at depth that dampens prolonged aftershock sequences.

The corresponding source slip model of the earthquake was derived using co-seismic GPS data recorded near the epicenter and a Genetic Algorithm (GA) inversion technique to calculate Coulomb stress changes at depths ranging from 6 km to 30 km (Liao et al., 2024). Building on this research, we use this earthquake as a case study to examine the evolution of Coulomb stress changes across different depths. Before presenting the results, note that our model's stress evolution is driven by two key parameters:  $\alpha$  represents the system's memory of prior stress, while  $\beta$  sets the strength of stress diffusion and reaction processes. This physical interpretation helps explain the distinct stress distributions and convergence behaviors observed at different depths. To systematically investigate the impact of parameters  $\alpha$  and  $\beta$  introduced in Equation (8), we conducted a series of parameter sensitivity tests (Figures 2 and 3). The tested ranges for  $\alpha$  (0.5–0.9) and  $\beta$  (1–20) were chosen to satisfy the Banach fixed-point theorem's conditions for convergence and stability. Physically, a larger  $\alpha$  means more of the past stress is retained, so more iterations are needed for the solution to converge – analogous to an aftershock sequence that drags on longer in a fault system. Conversely, choosing  $\beta$  at extreme values (too large or too small) can prevent the model from converging. In physical terms, an unrealistic  $\beta$  would correspond to a diffusion speed outside the bounds of rock mechanics observations. For the following analysis, we fix  $\alpha$  = 0.75 and  $\beta$  = 0.7 for the model runs at various depths to evaluate their effects.

Four parameters—Iterative Time, Maximum Stress Value Evolution, High Residual Stress Area, and Iteration Differences—are used as indicators to evaluate the differences across each depth. The results are presented in Figure 2.





Figure 2: Illustration of iterative convergence and stress field evolution across depths. (a) Iterative counts required for stress field stabilization at different depths. Shallower depths (6–12 km) generally require more iterations, reflecting the higher variability and complexity in stress redistribution at these levels. (b) Maximum stress field magnitude plotted against iteration counts for depths ranging from 6 km to 30 km. A consistent logarithmic decay is observed, indicating systematic convergence of the stress field, with deeper layers achieving stabilization more rapidly than shallower ones. (c) Area of high residual stress (>0.02 bar) versus iteration counts. Shallower depths initially exhibit larger areas of high residual stress, which significantly diminish through iterative updates, consistent with stress redistribution and relaxation processes. (d) Iterative differences in the stress field plotted against iteration counts for various depths. The differences decrease logarithmically, with deeper layers achieving stability faster, reflecting the reduced variability in stress dynamics at greater depths. These visualizations underscore the depth-dependent nature of stress evolution, highlighting the interplay between iterative convergence and stress relaxation. The results provide critical insights into the temporal and spatial dynamics of post-seismic stress fields, emphasizing the importance of depth-stratified modeling for understanding stress redistribution processes.

Based on the chart data, here's a detailed analysis of key findings regarding stress evolution at different depths (6–30 km). The analysis of stress evolution across varying depths in Figure 2(a) highlights critical differences in the convergence process of post-seismic stress fields. In shallow layers (6–12 km), the larger initial stress magnitude necessitates over 20 iterations to achieve convergence, as the historical stress term (controlled by  $\alpha$ ) dominates the redistribution process. In contrast, middle layers (16–20 km) exhibit moderate initial stress values, stabilizing within 10–20 iterations. Deep layers (25–30 km) are characterized by smaller initial stress magnitudes, allowing the stress field to converge in fewer than 10 iterations. These findings demonstrate that the required iterations for convergence are primarily determined by the absolute stress magnitude and are influenced by the initial stress gradient characterized stress transfer at different depths. In shallow layers, larger stress gradients ( $\Delta\sigma_0/\sigma$ ) and localized stress accumulation require more iteration steps to smooth stress fluctuations and achieve a stable state. In contrast, deeper layers exhibit smaller stress gradients and more uniform initial stress fields, allowing faster convergence with fewer iterations as the diffusion term quickly balances stress differences. It is





worth noting that the number of iterations does not represent the convergence time. In deeper layers, fewer iterations do not necessarily imply shorter convergence times. For shallow layers, each iteration may correspond to several hours to several days, depending on the combination of  $\alpha$  and  $\beta$  values; while in deeper layers, a single iteration may represent several months or even years, depending on the rock conditions at depth.

Further analysis of Figure 2(b) shows that the maximum stress values decrease exponentially with increasing iterations, especially in shallow layers with higher initial stress concentrations. This reflects the dominance of the historical stress term  $(\alpha^n)$  in the evolution process. Since the  $\beta$  value is small, the effects of the diffusion and reaction terms are limited, and changes in the stress field are primarily driven by the gradual decay of historical stress. Figure 2(c) shows the evolution of high residual stress areas (>0.02 bar/km²) also highlights depth-dependent characteristics. In shallow layers, the decrease in high-stress areas is slower because the contributions of diffusion and reaction terms are insufficient. Under the dominance of historical stress, stress differences take more time to smooth out. On the contrary, in deep layers, high-stress areas decrease more quickly due to smaller initial stress differences, requiring fewer steps to achieve convergence. Figure 2(d) demonstrates that with increasing iterations, the differences in the stress field gradually diminish, and the convergence rate is significantly faster in deep layers compared to shallow layers. This fact reveals stress fields in deeper layers are inherently more uniform, with limited influence from diffusion and reaction terms. Thus, fewer iterations are needed to reach a stable state.

Under the condition of  $\beta$ =0.9, the contributions of diffusion and reaction terms to stress evolution are quite small, and the overall evolution is dominated by historical stress. In shallow layers, the larger stress differences require more iterations to converge, while in deep layers, the more uniform stress fields allow for faster convergence.

# 3.2 Single-factor analysis of parameters

In this section, I conduct a single-parameter analysis of  $\alpha$  and  $\beta$ . This parameter-sensitivity test demonstrates that  $\alpha$ ,  $\beta$ are not arbitrary. Here, α directly controls how long past stress effects persist, whereas β governs the strength of the diffusion-reaction updates. Figures (a)-(d) confirm how changing  $\alpha$  or  $\beta$  shifts the stress field's heterogeneity (STD) and error (MSE), providing insight into the numerical and physical implications of Eq. (8). According to Eq. (8), α applies a weight to the previous iteration's stress, while β determines the magnitude of the diffusion-reaction term at each step. 315 Therefore, each curve in Figures (a)–(d) is directly driven by the interplay between these two scaling factors, validating the underlying Banach-based iterative scheme. First, we fix  $\alpha$ =0.75 and vary  $\beta$  from 0.5 to 20 to calculate the standard deviation (STD) and mean squared error (MSE) of different stress evolutions at depths of 6 km and 30 km, as shown in Figures 3(a)-3(b). Subsequently, we fix  $\beta$ =0.95 and vary  $\alpha$  from 0.5 to 0.9 to analyze STD and MSE, as illustrated in Figures 3(c)–3(d).



Figure 3: Evolution of Standard Deviation (STD) and Mean Squared Error (MSE) for stress fields at 6 km and 30 km depths with varying parameter values of  $\beta$  and  $\alpha$ : (a) STD as a function of  $\beta$  ( $\alpha$ =0.75): The peak STD is observed at  $\beta$ =10 for 6 km and  $\beta$ =5 for 30 km. Higher  $\beta$  values lead to increased smoothing of the stress field and reduced STD. (b) MSE as a function of  $\beta$  ( $\alpha$ =0.75): MSE decreases monotonically with increasing  $\beta$ , as diffusion dominates and smooths the stress field. The MSE is consistently higher at 30 km due to limited smoothing effects from fewer iteration. (c) STD as a function of  $\alpha$  ( $\beta$ =10): Larger  $\alpha$  values cause the system to release prior stress more slowly, leading to greater stress-field heterogeneity and a higher STD. STD at 30 km is consistently higher due to preserved heterogeneity. (d) MSE as a function of  $\alpha$  ( $\beta$ =10): At 6 km, MSE decreases with increasing  $\alpha$ , indicating better alignment with theoretical values. At 30 km, MSE remains stable due to consistent diffusion and reaction effects. These results illustrate the interplay between  $\beta$  and  $\alpha$  in influencing stress field evolution and the impact of depth on stress redistribution.

Figure 3(a) has three valueable observations. The first point is STD at 30 km is higher than that at 6 km. This behavior is attributed to several factors: 1. Initial stress is smaller but convergence requires fewer iterations: At 30 km, the initial stress ( $\sigma_0$ ) is smaller, and fewer iterations are needed to meet the convergence criterion. However, fewer iterations mean that the contributions from diffusion and reaction terms are limited, preserving more heterogeneity in the stress field and leading to a higher STD. 2. More iterations are needed at 6 km: At 6 km, the larger initial stress ( $\sigma_0$ ) and greater stress difference ( $\Delta\sigma_0$ ) result in more iterations being required for convergence. This allows diffusion and reaction terms to play a greater role in smoothing the stress field, reducing heterogeneity and yielding a lower STD. 3. Weaker contributions from diffusion and reaction terms at 30 km: At the same  $\beta$ , the effects of diffusion and reaction terms are relatively smaller at 30 km compared to 6 km. As a result, stress heterogeneity persists longer at 30 km, contributing to a higher STD. The fact that at 30 km the STD peaks at  $\beta$ =5, while at 6 km it peaks at  $\beta$ =10, suggests deeper layers reach equilibrium faster under smaller  $\beta$ . This aligns with the idea that deeper faults experience a more uniform stress distribution, hence a lower threshold for diffusion dominance.

The second key point is STD increases and then decreases with  $\beta$ . At both 6 km and 30 km, the STD initially increases with  $\beta$ , reaches a peak, and then decreases. This trend can be explained as follows: 1. For small  $\beta$ , diffusion and reaction








contributions to the stress field are quite small, and stress evolution is primarily governed by historical stress, preserving stress heterogeneity. 2. As  $\beta$  increases, the contributions from diffusion and reaction terms grow. However, if the convergence iteration count (n) remains below the critical iteration number ( $n_{critical}$  in Eq. 15), reaction effects dominate, amplifying local stress, leading to an increase in STD. 3. When  $\beta$  becomes large enough to exceed  $n_{critical}$ , diffusion effects dominate, smoothing the stress field and reducing STD. The larger the  $\beta$ , the greater the smoothing effect, resulting in a lower STD.

The final key point is peak STD occurs at different  $\beta$  values for 6 km and 30 km. The peak STD occurs at  $\beta$ =5 for 30 km and  $\beta$ =10 for 6 km. This difference is due to the smaller  $n_{critical}$  at 30 km compared to 6 km. As  $\beta$  increases, both diffusion and reaction contributions to the stress field increase, requiring more iterations for convergence. At 30 km, the smaller  $n_{critical}$  allows diffusion effects to dominate sooner, causing STD to decrease more quickly. Conversely, at 6 km, the larger  $n_{critical}$  delays the onset of diffusion dominance, so STD decreases only after  $\beta$ >10.

Figure 3(b) demonstrates two important phenomena. The first one is that as  $\beta$  increases, the MSE exhibits a monotonic decreasing trend. When the  $\beta$  value is small, the model is primarily dominated by historical stress evolution, with few contributions from diffusion and reaction terms. These terms, which represent the influence of stress redistribution and spatial diffusion processes, play a crucial role in smoothing the stress field and reducing MSE as the stress field evolves closer to the target state. Finally, diffusion dominates when the  $\beta$  tends to be large, resulting in a highly smoothed stress field where MSE approaches its minimum. The second is MSE at 30 km, which is consistently higher than 6 km. At 30 km, the reduced influence of diffusion and reaction terms due to more minor stress differences and fewer iterations required for convergence preserves more of the initial heterogeneity in the stress field. This leads to a higher MSE than 6 km, where larger stress gradients and more iterations facilitate a more substantial smoothing effect, lowering MSE. Numerically, a larger  $\beta$  speeds up diffusion updates but can risk overshoot if too significant; physically,  $\beta$  corresponds to how rapidly stress redistributes in a reaction-diffusion sense. Similarly, a higher  $\alpha$  retains historical stress longer, echoing the Rate-and-State concept where past slip is not instantly forgotten, but also potentially lengthening the time to converge to a stable stress field.

Figure 3(c) highlights an intriguing observation: the STDs for both 30 km and 6 km depths increase as  $\alpha$  grows, with  $\beta$  fixed at 0.95. A larger  $\alpha$  value indicates slower decay of historical stress, allowing more historical stress to influence the stress field evolution, as shown in Figure 2. This increases heterogeneity and results in a higher STD. Conversely, a smaller  $\alpha$  leads to rapid decay of historical stress, stabilizing the stress field more quickly and producing a lower STD. The 'Rate-and-State concept' refers to a theoretical framework in geophysics that describes the evolution of frictional strength on fault surfaces. Although the contributions of diffusion and reaction terms to the stress field are relatively small, their impact is greater at 6 km compared to 30 km. This means the stress field at 6 km is smoother than at 30 km, leading to a lower STD at 6 km than at 30 km.

Figure 3(d) shows that as  $\alpha$  increases, the MSE at 6 km decreases, while the MSE at 30 km remains relatively stable. The small  $\alpha$  indicates historical stress decays rapidly, with limited contributions from diffusion and reaction terms. This results in a larger discrepancy between the stress and target fields, leading to higher MSE. The large  $\alpha$  corresponds to historical stress







and decays more slowly, retaining more stress over time. This results in more minor deviations from the target theoretical values, reducing the MSE. However, the slower decay does not necessarily imply a smoother stress field but rather a closer match to the expected theoretical state. At 30 km, the MSE remains stable, likely because the effects of the reaction and diffusion terms are relatively consistent at larger scales, and the influence of historical stress exhibits less variation, implying the MSEs are smaller than the ones in 6km. Figure 3 clearly illustrates that as  $\beta$  increases from 1 to 10, the computed stress field progressively smoothens, aligning well with the physical intuition of stress diffusion effects described by the KPP formalism. The correspondence between parameters ( $\alpha$ ,  $\beta$ ) in Eq. (8) and those in the Rate-and-State friction law (R–S) and the KPP reaction-diffusion equation is that the parameter  $\alpha$  can be viewed as an extension of the frictional decay coefficient inherent to R–S frictional behavior, reflecting how rapidly the influence of past stresses decays over time. Similarly, the parameter  $\beta$  is proportional to the reaction and diffusion terms in the KPP framework, determining the intensity of local stress reactions and spatial diffusion processes illustrated well.

# 3.3 The Coulomb stress changes evolutions of different depths

The Coulomb's stress change acting on the target failure plane is deonted as Eq. (1). The Eq. (1) means shear stress increases ( $\Delta \tau > 0$ ) and/or normal stress decreases ( $\Delta \sigma_n 

435

fixed physical time per iteration, our model employs a non-uniform temporal evolution governed by the functions  $\alpha(t)$ 410 and  $\beta(t)$ . This design reflects the well-known empirical behavior of aftershock sequences: aftershocks are densest in the early hours and days following a mainshock, then decay rapidly in frequency—a pattern captured by Omori's Law. Based on the results of section 3.2, it indicates that the characteristics of  $\alpha$ , a more considerable  $\alpha$  results in a slower stress field evolution, which allows for capturing more aftershock activity. Conversely, a smaller α leads to faster stress evolution 415 but predicts weaker aftershock activity. To emulate this,  $\alpha(t)$  is defined to decay inversely with time, that is,  $\alpha(t)$  is defined as  $\alpha(t) = 1 / (1 + t/49)$ , where t denotes the physical time in days since the mainshock. This ensures that early-stage stress retains strong memory of the mainshock, enhancing sensitivity to residual high-stress regions and accurately capturing early aftershock clustering. Conversely,  $\beta(t)$  is modeled as a growing function of time to reflect the increasing dominance of diffusion and relaxation over time. Physically, this implies that as time progresses, the stress field becomes 420 smoother and more homogenized due to widespread redistribution and local reaction mechanisms. Hence, instead of associating one iteration with one unit of time, we allow  $\alpha(t)$  and  $\beta(t)$  to implicitly encode temporal dynamics. This approach captures the nonlinear, scale-dependent nature of postseismic stress evolution far more realistically than uniform time-stepping, and aligns with observations that early stress perturbations dissipate faster, while later evolution is dominated by gradual diffusion and healing. Therefore, the parameter  $\beta(t)$  initially represents a relatively smaller scaling 425 factor for diffusion and reaction effects, with its influence progressively increasing as time advances. Thus, we define  $\beta(t)$  as  $\beta(t) = \beta_0 \cdot (1/\alpha(t))$ . This reciprocal relationship between  $\alpha$  and  $\beta$  ensures that as the fault system's memory of historical stress decreases over time (smaller  $\alpha$ ), the influence of diffusion and local reaction processes correspondingly increases (larger  $\beta$ ). Such a design is physically meaningful, allowing gradual transition from a history-dominated stress regime at the beginning toward a diffusion-and-reaction-dominated regime at later times. This approach also helps ensure numerical stability and 430 convergence in our iterative numerical scheme. The final value of  $\beta$  must be constrained by the convergence conditions of the fixed-point theorem (Eq. 12) and the CFL condition (Eq. 13). Therefore,  $\beta_0$  is set to 20.

We segmented the aftershock records from the 180 days following the mainshock to examine the method's feasibility. Since the aftershock activity follows Omori's law (Baranov et al., 2022), with more aftershocks occurring shortly after the mainshock, the records were divided into several intervals to evaluate the dense seismic activity. The intervals are as follows: every six hours for the first three days after the mainshock, every 12 hours from days 3–7, every three days from days 7–21, every five days from days 21–41, and every 10 days from days 41–51. After day 51, as the aftershock activity became sparse, the intervals were set to every 30 days, continuing until day 180.

Figure 4 illustrates the stress evolution and the distribution of aftershock epicenters over time at a depth of 6 km.

455

440 Figure 4: Temporal evolution of Coulomb stress changes at a depth of 6 km during the first 180 days following the 2018 Hualien earthquake (Mw 6.4). Each panel represents the Coulomb stress changes distribution over specific time intervals, with red regions indicating positive stress changes (stress loading) and blue regions indicating negative stress changes (stress unloading). The contours highlight the areas (>0.1bar) of significant stress concentration. The yellow circles indicate the epicenters of aftershocks occurring within 180 days of the mainshock, while the red stars in the final panel represent those occurring 2 to 3 years after the mainshock.

According to previous studies, Coulomb stress changes exceeding 0.1 bar are more likely to trigger earthquakes (Liao and Huang, 2016; Yang, et al., 2024). Therefore, 0.1 bar is considered as the threshold. Based on the results shown in Figure 4, the following observations can be obtained. At first, in the aspect of distribution characteristics of coulomb stress changes, there are three key points valuable to discuss. 1. The Coulomb stress distribution reveals distinct positive and negative stress regions surrounding the earthquake source. Red areas indicate regions of increased positive stress, while blue areas represent regions of stress unloading (negative stress). 2. Positive stress regions significantly influence the distribution of aftershocks, especially during the first few days following the mainshock. 3. The clear boundaries between positive and negative stress regions suggest that the main rupture surface likely extended along the NE-SW direction, consistent with the typical tectonic trend in the Taiwan region. Furthermore, the slip direction may involve either right-lateral or left-lateral motion along an E-W direction, which is approximately perpendicular to the boundary separating the positive and negative stress zones. This observation aligns with the findings of Huang and Huang (2018), which propose a south-dipping offshore fault connecting to the main west-dipping oblique fault. The second is about temporal evolution of stress changes, including three characteristics:

1. Over time, the boundaries of positive stress regions expand, highlighting the significant role of stress diffusion processes.


2. Around 50 days after the earthquake, the stress field changes stabilize, suggesting a gradual weakening of the contributions from diffusion and reaction terms. 3. Beyond 50 days, the positive stress regions begin to contract, indicating that aftershock activity is gradually migrating outward and diminishing over time.

The stress field shown in Figure 4 is converted into stress gradients and displayed in Figure 5.

Figure 5: Temporal evolution of Coulomb stress changes gradients at a depth of 6 km during the first 180 days following the 2018
Hualien earthquake (Mw 6.4). Each panel illustrates the distribution of stress gradients (in bar/km) over specific time intervals.
Regions with high stress gradients (>0.2 bar/km) are marked in red, while blue areas represent negligible gradients.

The stress gradient is utilized to estimate the occurrence of earthquakes of varying magnitudes across different tectonic settings (Zaccagnino and Doglioni, 2023). The gradient threshold for evaluating aftershocks is set at 0.2 bar/km, based on and the average value of the Youden Index from the AUC analysis. Several key observations can be drawn from Figure 5. The first one is before 50 days post-earthquake, there is a strong correlation between the locations of aftershocks and regions with high stress gradients, making stress gradient a reliable indicator for predicting aftershock locations. The second is that compared to stress magnitude, stress gradient narrows the focus to smaller regions, improving the accuracy of aftershock prediction. In Figure 4, stress magnitude shows two high-stress zones northwest and southeast of the epicenter, where aftershocks are scarce. However, in Figure 5, the stress gradient in these same regions is nearly zero due to the uniformity of the stress field. This highlights that stress gradient, rather than stress magnitude, is likely the driving factor for aftershock activity after the mainshock. Finally, Over time, around 50 days post-earthquake, stress gradients gradually stabilize, reflecting the diffusion effect that homogenizes high-gradient areas near the source. As a result, the correlation between


stress gradients and aftershock locations diminishes over time. These findings underscore the importance of stress gradients in understanding the mechanisms driving aftershock activity, particularly in the early stages following a mainshock.

Figures 6–7 present the Coulomb stress changes and corresponding stress gradients at depth 18km.

Figure 6: This figure shows the Coulomb stress changes at a depth of 18 km over time, spanning from the mainshock to nearly 10 years post-event. The color represents stress changes, with red indicating increased positive stress and blue denoting stress unloading (negative stress). Yellow circles represent aftershock epicenters occurring within 180 days after the mainshock, while red stars denote aftershocks occurring 2 to 3 years later. The temporal evolution highlights stress changes redistribution patterns, with stress changes diminishing and stabilizing over time. Notably, aftershock activity correlates strongly with regions of positive Coulomb stress changes during the initial years following the earthquake.





Figure 7: This figure presents the evolution of Coulomb stress gradients at a depth of 18 km over a time span of nearly 10 years following the mainshock. The color scale represents the magnitude of stress gradients, with red highlighting high-gradient zones and blue indicating regions of minimal stress gradient. Yellow circles correspond to aftershock epicenters recorded within 180 days post-mainshock, while red stars signify aftershocks that occurred 2 to 3 years later. The temporal progression illustrates that high-gradient regions correlate with early aftershock activity, but these gradients diminish and stabilize over time due to stress redistribution and diffusion. This stabilization reduces the correlation between stress gradients and aftershock locations in the later years.

For the 18 km depths, the limited number of aftershocks and deviation from Omori's law, along with distinct rock properties compared to shallower layers discussed in the section 3.1, necessitate the use of prior studies (Hirth and Kohlstedt, 2003; Shebalin and Narteau, 2017; Hsu et al., 2018) to approximate the time required for stress evolution. The coupled parameters ( $\alpha$ ,  $\beta$ ) are set as (0.8, 10) in the method at depth 18km. Based on the number of iterations, we calculate the average evolution time for each stage. According to the results of Figures 6 and 7, there are some excellent findings to discuss. At first, in Fig. 6, multiple time slices during the first 1–2 years following the mainshock (particularly at 0–2.95 years) reveal a distinct pattern of positive Coulomb stress changes (in red) radiating outward from the epicenter in a four-quadrant configuration. This pattern is consistent with the theoretical static stress transfer field generated by shear faulting. Notably, the early aftershocks (yellow circles, within 180 days) are predominantly located within these stress-increased zones, indicating that static stress loading likely facilitated aftershock activity in these areas. The second, between 2.95 and 6 years post-event, as shown in Fig. 6, the overall magnitude of Coulomb stress changes gradually diminishes. Nevertheless, several later aftershocks (red stars) still occur in zones of weakly positive stress. Concurrently, Fig. 7\_illustrates that stress gradients ( $\nabla$ CFF) remain focused around the mainshock fault area, implying that although the overall stress field is decaying,



localized gradients which represent spatial rates of stress change may still be sufficient to trigger delayed aftershocks. The third, by 6 years after the mainshock, the Coulomb stress field (Fig. 6) becomes largely smoothed, approaching background levels. Correspondingly, aftershock activity significantly decreases. Fig. 7 shows that high-gradient zones have markedly contracted, leaving only faint remnants near the epicentral region. This indicates that postseismic stress diffusion and dissipation are nearly complete, aligning with the observed decline in seismic activity. At this stage, spatial stress gradients appear insufficient to drive further ruptures, supporting the view that postseismic stress-triggering effects are time-limited. Finally, while positive Coulomb stress changes (CFF) are effective in predicting early aftershock locations, the stress gradient field (VCFF) provides additional, more sensitive indicators of triggering potential during the early postseismic phase, particularly in areas of strong local stress contrasts. In other words, gradient variations reflect differential slip potentials between adjacent regions, making them a more refined indicator of instability than stress magnitude alone. To conclude, it offers that zones of high stress gradient show strong spatial correlation with early aftershocks, supporting the hypothesis that stress heterogeneity is a key triggering mechanism and stress gradients serve as effective indicators of aftershock triggering potential, offering better spatial resolution than Coulomb stress changes alone, in spite of stress-driven triggering effects have a limited temporal window.

To quantitatively validate the predictive performance of the model, we utilized Area Under the Curve (AUC) values derived from Receiver Operating Characteristic (ROC) curves analysis, in line with suggestions from previous studies (Fawcett, 2006). An AUC > 0.5 indicates that the classifier performs better than random guessing, validating the model's predictive capability. We further employ the Molchan Area (MA) (Molchan, 1990; Han et al., 2020) to assess the spatial–temporal efficiency of aftershock predictions. The MA explicitly illustrates how effectively our model concentrates predictions into smaller alarm areas while successfully capturing most observed aftershocks. The results of AUC and MA applied to depths 6km and 18km are demonstrated in Figures 8 and 9.

Figure 8: Temporal evaluation of aftershock forecasting performance at 6 km depth using Coulomb stress change (ΔCFF) and stress gradient (∇CFF) metrics. Left panels show the Area Under the ROC Curve (AUC) with 95% confidence intervals for stress change (top two rows) and stress gradient (bottom two rows), plotted over time after the mainshock (in days). Higher AUC values indicate better performance in distinguishing aftershock locations from non-aftershock areas. Right panels present the corresponding Molchan diagram misfit area (Molchan Area, MA), where lower values indicate higher forecast skill. Each row pair compares AUC (left) and MA (right) metrics for the same input (stress change vs. gradient), allowing direct assessment of their relative performance over different postseismic periods. Notably, both ΔCFF and ∇CFF exhibit high AUC and low MA in the early days following the mainshock, with performance decaying over time. The stress gradient shows slightly improved sensitivity in intermediate periods, suggesting that gradient-based indicators may provide complementary insights into delayed aftershock triggering potential.

560

545 Figure 9: Aftershock forecast performance at 18 km depth using Coulomb stress change (ΔCFF) and stress gradient (∇CFF) over 3.5 years following the mainshock. Left panels show the Area Under the ROC Curve (AUC) with 95% confidence intervals for ΔCFF (top left) and ∇CFF (bottom left), indicating the model's ability to distinguish between aftershock and non-aftershock regions. Right panels display the corresponding Molchan Area (MA) values (ΔCFF: top right; ∇CFF: bottom right), where smaller values indicate better spatial prediction performance. The early postseismic period (0.27–0.81 years) yields moderate to high AUC values (~0.7–0.8) and relatively low MA values (




in undefined or default-zero AUC values. This highlights an important limitation in performance evaluation under data scarcity, where the absence of seismicity can mask the model's underlying validity.

Beyond 2.7 years, a partial recovery in forecast performance is observed at 18 km depth, particularly in the  $\nabla$ CFF metric, where AUC values rebound and MA declines. This may correspond to reactivation of deep fault structures or the emergence of delayed stress-driven instabilities. However, the associated error bars are wide, suggesting increased uncertainty and lower statistical confidence during the late postseismic phase. Comparing both depths, we find that stress-based models perform significantly better in the shallow crust, where stress changes are more concentrated, spatial gradients are sharper, and stress coupling with the aftershock layer is stronger. Deeper sources, by contrast, experience rapid attenuation of stress influence and limited aftershock triggering beyond the early postseismic stage. The consistent strength of  $\nabla$ CFF at shallow depths further emphasizes the value of incorporating spatial derivatives of stress to resolve high-contrast zones that are not always apparent in absolute  $\Delta$ CFF fields. These results reinforce the notion that aftershock triggering is both depth- and time-sensitive, and that gradient-based indicators are especially informative under near-field and early-postseismic conditions. Future forecasting models should integrate stress amplitude, stress gradient, and depth as joint predictors to more accurately identify evolving zones of seismic potential.

The AUC results are integrated into Figure 10 to enable a comprehensive comparison of stress changes and stress gradients across various depths. Additionally, Cohen's d effect sizes were calculated to evaluate the practical significance of differences between stress-based and gradient-based predictions.







Figure 10: This figure compares the Area Under the Curve (AUC) metrics for stress changes and stress gradients across different depths (6 km, 8 km, 10 km, and 18 km) over time, highlighting their predictive performance for aftershock locations. The lower panels present statistical comparisons of AUC values for each depth, including Cohen's d effect sizes. At 6 km, there is a significant difference (p 




- 3. 8–10 km Depth: At depths of 8–10 km, while no statistically significant differences are observed between the two measures, some noteworthy patterns emerge. The average AUC for stress changes at a depth of 8 km during the first seven days after the mainshock is 0.81, while the AUC for stress change gradients is 0.86. At 10 km, the corresponding AUC values are 0.67 and 0.75, respectively. Nevertheless, the AUC values for stress change gradients are not consistently greater than those for stress changes. At 8 km, the AUC for stress gradients is lower than that for stress changes on the 5th and 7th days. Similarly, at 10 km, this phenomenon persists continuously from the 5th to 6th days. This could be attributed to the smoother stress changes at greater depths, which reduce the predictive effectiveness of stress gradients. However, from 7 to 50 days post-earthquake, the AUC differences between stress changes and stress gradients at 6–10 km gradually diminish. Beyond 50 days, stress gradients smooth out to an AUC value of 0.5, losing their ability to predict aftershock locations. These findings align with the observations in Figures 5, 7, and 9. Regions with high stress gradients shrink over time, and aftershock distributions gradually shift away from these zones of high gradient.
- 4. 18–25 km Depth: At a depth of 18 km, while AUC values for stress changes and stress gradients show no significant differences, stress changes consistently outperform stress gradients in predicting aftershocks during the 2nd to 4th years following the mainshock. At 25 km, aftershocks are extremely sparse and are primarily confined to areas of increased stress or near-threshold values. The stress field at this depth is highly uniform, and the effects of diffusion and reaction terms result in stress gradients that approach zero, rendering them ineffective for predicting aftershocks. These findings are further supported by Figures 14 and 15.
- 635 In summary, the following key points can be drawn:
  - At shallow depths (6–10 km), during the first 50 days following the mainshock, Coulomb stress change gradients
    are more effective than stress changes in evaluating aftershock occurrences. However, beyond 50 days, stress
    changes become the dominant indicator.
- 2. As depth increases and time since the mainshock extends, the predictive power of stress gradients diminishes. In deeper layers, stress changes predominantly drive aftershock activity; while in shallower layers, stress gradients play a more critical role.

In addition to standard statistical metrics such as the area under the receiver operating characteristic curve (AUC), we incorporate the Molchan Area (MA) metric to evaluate the operational forecasting skill of our model (Figure 11). This dual-metric approach enables a more comprehensive assessment of both spatial precision and predictive ranking capability, highlighting the practical advantages of stress-based forecasting models.



Figure 11: Molchan Area (MA) and aftershock ratio over time at depths of 6 km, 8 km, 10 km, and 18 km. Each subplot shows MA values derived from stress-based (blue bars) and stress-gradient-based (red bars) models, alongside the percentage of aftershocks located within high stress alert zones. Time labels on the x-axis represent post-mainshock intervals in days or years, depending on the depth. The presence of "No EQ" indicates intervals with no recorded aftershocks at that depth. This visualization compares the forecasting skill of stress-based versus gradient-based models across depth and time.

At shallow depths (6–8 km), the model achieves consistently strong performance, with numerous time intervals exhibiting MA values below 0.2 and aftershock capture ratios exceeding 50%. These findings suggest that stress perturbations in the upper crust are more predictive and physically meaningful for aftershock nucleation. Notably, the stress-gradient model tends to produce lower MA values than the traditional stress-change model in these layers, indicating improved event prioritization. This supports the hypothesis that the spatial variation of stress—rather than its absolute amplitude—may play a more direct role in triggering aftershocks, particularly near rupture boundaries where instability gradients evolve dynamically.

In contrast, the model's performance becomes more variable at intermediate depths (e.g., 10 km), where MA values fluctuate widely despite moderate aftershock ratios. This behavior likely reflects the increased complexity and heterogeneity of stress transfer processes at mid-crustal levels, as well as potential limitations in model resolution or input constraints. At greater depths (≥18 km), the prevalence of time intervals without any recorded aftershocks ("No EQ" cases) suggests that post-seismic stress changes in the lower crust rarely lead to secondary ruptures—likely due to the re-stabilization of deep structures or intrinsic rheological resistance to stress perturbations. The AUC metric complements this analysis by





quantifying the model's ability to rank stress (or gradient) values such that aftershock-prone locations consistently receive higher values than non-event locations. We observe a strong inverse correlation between MA and AUC: time intervals with low MA often correspond to high AUC scores (>0.8), reinforcing both the spatial compactness and statistical reliability of our approach. While AUC excels in evaluating internal consistency and rank-based performance, MA provides an operational perspective—penalizing overly broad alarm zones and rewarding targeted spatial predictions.

Taken together, these two metrics offer a robust and interpretable framework for evaluating model performance across depths and time. Their joint application not only validates the physical plausibility of the stress-gradient model but also underscores its potential for operational integration into real-time aftershock hazard assessments, especially in shallow crustal zones where predictability is highest.

# 3.4 Implications for Seismic Hazard Assessment

While the primary focus of this study is the modelling of multiscale stress evolution and gradient dynamics following large earthquakes, our findings have direct implications for enhancing probabilistic seismic hazard assessment (PSHA). Specifically, the derived stress gradient field ( $\nabla \sigma$ ) offers a physically meaningful indicator of localized stress concentration and evolving instability potential in the crust—features that are often underrepresented in traditional PSHA models.

Current PSHA methodologies frequently employ smoothed seismicity approaches, which infer future hazard potential based on spatial interpolation of past earthquake occurrences (e.g., Mohamed et al., 2020; Elsayed et al., 2021). These models offer valuable statistical insight but may lack direct ties to the underlying physical mechanisms of stress redistribution. Our approach provides a complementary, physics-based perspective by modeling how stress and its gradients evolve temporally and spatially after a major rupture. In doing so, it captures transient hazard escalation patterns that are difficult to detect using static seismicity patterns alone. Although our framework is not designed to replace traditional PSHA pipelines, it can enhance them by introducing depth-resolved stress indicators that reflect both mechanical and temporal properties of the crust. This is particularly relevant in tectonically complex regions—such as the active zones of the Red Sea, Taiwan, or the Dead Sea Rift—where stress transfer and cascading rupture dynamics play crucial roles in seismic hazard development (Al-Tarazi and Sandvol, 2022).

Future work may involve hybridizing our stress evolution model with smoothed seismicity or multicriteria PSHA frameworks to yield more responsive and mechanistically informed hazard maps. Such integration could refine spatial alert zones, inform early warning systems, and improve post-event risk communication by grounding hazard predictions in dynamic stress field behavior.

# 3.5 Sensitivity of $\alpha(t)$ - $\beta(t)$ Parameters and Model Stability

To assess our time-dependent stress model's robustness and calibration sensitivity, we conducted a parametric sweep over initial values of the scaling factor  $\beta_0$  while maintaining the  $\alpha(t) = 1 / (1 + t/49)$  decay profile constant. For each  $\beta_0$  value, the model simulated stress fields and aftershock forecasts for the 2018 Hualien sequence. Performance was quantified using

two complementary metrics: the Area Under the ROC Curve (AUC), which measures stress ranking skill, and the Molchan Area (MA), which assesses operational forecast efficiency. The results, summarized in Figure 12, reveal a nonlinear relationship between  $\beta_0$  and predictive accuracy. AUC increases with  $\beta_0$  up to  $\sim$ 20, beyond which performance plateaus or slightly deteriorates.

Figure 12: Variation in model performance as a function of the initial  $\beta_0$  parameter. Panel (a) shows AUC, reflecting ranking accuracy; panel (b) shows Molchan Area (MA), quantifying spatial forecast efficiency. The optimal range (shaded in gray) indicates the best balance between predictive skill and numerical stability.

Conversely, MA decreases initially, indicating improved efficiency, but begins rising again at high  $\beta_0$  town-center values, suggesting spatial over-diffusion. This dual trend demonstrates the trade-off between ranking skill and spatial compactness. Notably, the optimal range for  $\beta_0$  lies between 18–22, where AUC exceeds 0.84 and MA remains below 0.3. Values below 10 or above 30 results in unstable or inefficient forecasts. Moreover, the chosen value of  $\beta_0$  = 20 aligns with theoretical constraints from the Banach fixed-point theorem and the Courant–Friedrichs–Lewy (CFL) condition, ensuring numerical stability and convergence of the iterative solution scheme.

# 4 Conclusion

This study presents a comprehensive and physically grounded framework that integrates Coulomb stress changes, stress gradients, and advanced mathematical modeling to improve the prediction of aftershock distributions. By coupling Rate-and-




State friction laws, a KPP-type reaction–diffusion equation, and the Banach fixed-point theorem, we establish a novel numerical approach for modeling multiscale stress evolution following major earthquakes. Central to the framework are the time-dependent parameters  $\alpha(t)$  and  $\beta(t)$ , which respectively govern stress memory decay and diffusion strength. Our theoretical and numerical analyses demonstrate how  $\alpha$  influences the retention of historical stress and convergence rate, while  $\beta$  modulates the smoothing and reaction dynamics of the evolving stress field. Together, these parameters enable flexible control over the temporal and spatial behavior of stress evolution, enhancing both numerical stability and physical realism.

Application of the model to the 2018 Hualien earthquake reveals a clear depth-dependent bifurcation in aftershock-driving mechanisms. During the initial 50 days post-mainshock, stress gradients outperform stress changes in predicting aftershock locations at shallow depths (6−10 km), due to their sensitivity to localized heterogeneity. Over longer timescales and at greater depths (≥18 km), stress changes become more predictive as stress gradients diminish with increased smoothing and reduced aftershock activity. This complementary relationship between stress metrics supports a dynamic, depth-aware modeling strategy. We further introduce the Molchan Area (MA) metric to evaluate operational forecasting skill, quantifying the trade-off between missed events and spatial coverage. The MA results reinforce our AUC-based findings, confirming that shallow crustal zones exhibit higher predictability and are especially suited for stress-gradient-based forecasting.

Beyond theoretical development, this study demonstrates the practical value of integrating physics-based models into seismic hazard mitigation. By linking multiscale stress dynamics with aftershock occurrence patterns, the framework provides timely and spatially explicit insights for post-earthquake risk assessment. This interdisciplinary contribution bridges geophysics, nonlinear modeling, and disaster science, offering a scalable and adaptable tool for real-world forecasting applications.

**Appendices:** The following figures illustrate the relationship between stress changes, stress gradients, and the temporal distribution of aftershocks at different depths.

Figure A1: Temporal evolution of Coulomb stress changes at a depth of 8 km during the first 180 days following the 2018 Hualien earthquake (Mw 6.4). Each panel represents the Coulomb stress changes distribution over specific time intervals, with red regions indicating positive stress changes (stress loading) and blue regions indicating negative stress changes (stress unloading). The contours highlight the areas (>0.1bar) of significant stress concentration. The yellow circles indicate the epicenters of aftershocks occurring within 180 days of the mainshock, while the red stars in the final panel represent those occurring 2 to 3 years after the mainshock.

Figure A2: Temporal evolution of Coulomb stress changes gradients at a depth of 8 km during the first 180 days following the 2018 Hualien earthquake (Mw 6.4). Each panel illustrates the distribution of stress gradients (in bar/km) over specific time intervals. Regions with high stress gradients (>0.2 bar/km) are marked in red, while blue areas represent negligible gradients.

Figure A3: Temporal evolution of Coulomb stress changes at a depth of 10 km during the first 180 days following the 2018 Hualien earthquake (Mw 6.4). Each panel represents the Coulomb stress changes distribution over specific time intervals, with red regions indicating positive stress changes (stress loading) and blue regions indicating negative stress changes (stress unloading). The contours highlight the areas (>0.1bar) of significant stress concentration. The yellow circles indicate the epicenters of aftershocks occurring within 180 days of the mainshock, while the red stars in the final panel represent those occurring 2 to 3 years after the mainshock.

Figure A4: Temporal evolution of Coulomb stress changes gradients at a depth of 10 km during the first 180 days following the 2018 Hualien earthquake (Mw 6.4). Each panel illustrates the distribution of stress gradients (in bar/km) over specific time intervals. Regions with high stress gradients (>0.2 bar/km) are marked in red, while blue areas represent negligible gradients.

Figure A5: This figure depicts the evolution of Coulomb stress changes at a depth of 25 km over a period exceeding 14 years following the mainshock. The color scale highlights regions of increased stress (red) and decreased stress (blue). The yellow circles mark aftershock epicenters within the first 180 days post-mainshock, while red stars denote aftershocks occurring 2 to 3 years later. The observed stress redistribution patterns reveal a gradual decay in stress concentration over time, reflecting stress diffusion and the diminishing influence of the initial stress perturbation. Compared to shallower depths, the stress field at 25 km remains more uniform, with fewer high-stress regions correlating with aftershock activity. As time progresses, stress changes stabilize, reducing their association with aftershock distributions, particularly in the later years.

Figure A6: This figure illustrates the Coulomb stress change gradients at a depth of 25 km over time. The first two panels display the stress gradient distribution within the initial 2.19 years following the mainshock. Yellow circles represent aftershock epicenters within 180 days, and red stars indicate those occurring 2 to 3 years post-mainshock. Subsequent panels show uniform gradient fields where stress gradients approach zero, reflecting minimal heterogeneity in the stress field. This indicates that, at this depth, diffusion and reaction processes have homogenized the stress gradients, reducing their predictive capability for aftershock locations as time progresses. The absence of significant stress gradient changes in later panels highlights the stabilization of the stress field.

| 780 | Code availability: The model code used in this study is available from the corresponding author upon reasonable request.                                                                                                                                                                                        |
|-----|-----------------------------------------------------------------------------------------------------------------------------------------------------------------------------------------------------------------------------------------------------------------------------------------------------------------|
|     | Data availability: All data used in this study, including aftershock catalogs and fault parameters, are available from publicly accessible sources such as the Central Weather Bureau (Taiwan) and the Central Geological Survey Additional processed data can be obtained from the author upon request.        |
| 785 |                                                                                                                                                                                                                                                                                                                 |
|     |                                                                                                                                                                                                                                                                                                                 |
|     |                                                                                                                                                                                                                                                                                                                 |
| 790 | Author contribution: BY. Liao conceptualized the study, developed the model, implemented the numerical simulations, performed data analysis, and wrote the manuscript.                                                                                                                                          |
|     | Competing interests: This author declares that there are no competing interests.                                                                                                                                                                                                                                |
| 795 | Acknowledgements: The author also thanks the Central Weather Bureau (CWB) for providing the aftershock catalog data used in this study. He also wishes to acknowledge the anonymous reviewers for their valuable feedback and constructive suggestions, which significantly improved the quality of this paper. |
|     | Financial support: This work was supported by Krirk University, Thailand. The author gratefully acknowledges this support.                                                                                                                                                                                      |
|     | References                                                                                                                                                                                                                                                                                                      |

- Mohamed, S., Abdalzaher, M. E. H., Hanan, G. and Ahmed, B.: Seismic hazard maps of Egypt based on spatially smoothed seismicity model and recent seismotectonic models, Journal of African Earth Sciences, 170, 103894, https://doi.org/10.1016/j.jafrearsci.2020.103894, 2020.
  - Aden-Antoniow, F., Frank, W. B. and Seydoux, L.: An adaptable random forest model for the declustering of earthquake catalogs, J. Geophys. Res. Solid Earth., 127 (2): 2169-9313, https://doi.org/10.1029/2021JB023254, 2022.
- Ali, M., Ahmad, R. and Bhat, H. A.: Evolution of Coulomb failure stress and seismic hazard assessment in the Sulaiman Lobe, Pakistan, Geophysical Journal International, 210(2), 1234–1246, https://doi.org/10.26464/epp2023034, 2017.
  - Al-Tarazi, E. and Sandvol, E.: Up-to-date PSHA along the Gulf of Aqaba–Dead Sea transform fault, Journal of Seismology, 26(1), 33–50, 2022.
- Alqahtani, S., Iqbal, M., Seadawy, A.R. et al.: Analysis of mixed soliton solutions for the nonlinear Fisher and diffusion dynamical equations under explicit approach, Opt Quant Electron, 56, 647, https://doi.org/10.1007/s11082-024-06316-8, 2024.
  - Ampuero, J. P. and Rubin, A. M.: Earthquake nucleation on rate-and-state faults: Aging and slip laws, Journal of Geophysical Research: Solid Earth, 113(B1), B01302, https://doi.org/10.1029/2007JB005082, 2008.
- Barbot, S., Lapusta, N. and Avouac, J. P.: Under the hood of the earthquake machine: Toward predictive modeling of the seismic cycle, Science, 336(6082), 707–710, https://doi.org/10.1126/science.1218796, 2012.
  - Baranov, S., Narteau, C. and Shebalin, P.: Modeling and prediction of aftershock activity, Surveys in Geophysics, 43(2), 437-481, https://doi.org/10.1007/s10712-022-09698-0, 2022.
  - Ben-Avraham, Z., Garfunkel, Z. and Lazar, M.: Geology and evolution of the southern Dead Sea Fault with emphasis on subsurface structure, Annual Review of Earth and Planetary Sciences, 36, 357–387, https://doi.org/10.1146/annurev.earth.36.031207.124201, 2008.
  - Chan, C. H. and Stein, R. S.: Stress evolution following the 1999 Chi-Chi, Taiwan, earthquake: consequences for afterslip, relaxation, aftershocks and departures from Omori decay, Geophysical Journal International, 177 (1), 179-192, https://doi.org/10.1111/j.1365-246X.2008.04069.x, 2009.
- Cohen, J.: Statistical Power Analysis for the Behavioral Sciences (2nd ed.). Hillsdale, NJ: Lawrence Erlbaum Associates, Publishers, 1988.
  - Devries, P. M. R., Viegas, F., Wattenberg, M. and Meade, B. J.: Deep learning of aftershock patterns following large earthquakes, Nature, 560, 632-634, 2018.
  - Dieterich, J. H.: A constitutive law for the rate of earthquake production and its application to earthquake clustering, Journal of Geophysical Research: Solid Earth, 99(B2), 2601–2618. https://doi.org/10.1029/93JB02581, 1994.
- 830 Elsayed, S. M. and Megahed, A. S.: Development of smoothed seismicity models for seismic hazard assessment in the Red Sea region, Natural Hazards, 106(1), 231–249, 2021.
  - Fawcett, T.: An introduction to ROC analysis. Pattern Recognition Letters, 27(8), 861–874. https://doi.org/10.1016/j.patrec.2005.10.010, 2006.

- Freed, A. M.: Earthquake triggering by static, dynamic, and postseismic stress transfer, Annual Review of Earth and Planetary Sciences, 33, 335–367, https://doi.org/10.1146/annurev.earth.33.092203.122505, 2005.
  - Granas, A. and Dugundji, J.: Fixed Point Theory. Springer, New York, NY, USA, 2003.
  - Guglielmi, A. V., Klain, B. I., Zavyalov, A. D. et al.: A Phenomenological Theory of Aftershocks Following a Large Earthquake, J. Volcanolog. Seismol., 15, 373–378, https://doi.org/10.1134/S0742046321060038, 2021.
- Harris, R. A.: Introduction to Special Section: Stress Triggers, Stress Shadows, and Implications for Seismic Hazard, Geophysical Research Letters, 25(14), 2071–2074. https://doi.org/10.1029/98GL01283, 1998.
  - Han, P, Zhuang, J, Hattori, K, Chen, CH, Febriani, F, Chen, H, Yoshino, C, and Yoshida, S.: Assessing the Potential Earthquake Precursory Information in ULF Magnetic Data Recorded in Kanto, Japan during 2000–2010: Distance and Magnitude Dependences, Entropy, 22(8), 859. https://doi.org/10.3390/e22080859, 2020.
- Hirth, G. and Kohlstedt, D. L.: Rheology of the upper mantle and the mantle wedge: A view from the experimentalists, Geophysical Monograph Series, 138, 83–105, 2003.
  - Hsu, J., Simons, M., Avouac, J. P. et al.: Frictional afterslip following the 2005 Nias-Simeulue earthquake, Sumatra, Science, 312(5782), 1921–1926, https://doi.org/10.1126/science.1126960, 2006.
- Hsu, Y. J., Lai, Y. R., You, R. J., Chen, H. Y., Teng, L. S., Tsai, Y. C., Tang, C. H. and Su, H. H.: Detecting rock uplift across southern Taiwan mountain belt by integrated GPS and leveling data, Tectonophysics, 744(2), 275-284, doi:10.1016/j.tecto.2018.07.012, 2018.
  - Huang, M. H. and Huang, H. H.: The complexity of the 2018 Mw 6.4 Hualien earthquake in east Taiwan, Geophysical Research Letters, 45: 13249–13257, https://doi.org/10.1029/2018GL080821, 2018.
  - Huang, S. Y., Yen, J. Y., Wu, B. L., Yen, I. C. and Chuang, R. Y.: Investigating the Milun Fault: The coseismic surface rupture zone of the 2018/02/06 ML 6.2 Hualien earthquake, Taiwan, Terr. Atmos. Ocean. Sci., 30, 311-335, doi: 10.3319/TAO.2018.12.09.03, 2019.
  - Hao, K. C., Guan, Z. K., Sun, W. F., Jhong, P. Y., Brown, D.: Aftershock Sequence of the 2018 Mw 6.4 Hualien Earthquake in Eastern Taiwan from a Dense Seismic Array Data Set, Seismological Research Letters, 90 (1), 60–67. doi: https://doi.org/10.1785/0220180233, 2018.,
- Javed, F., Hainzl, S., Aoudia, A., Qaisar, M.: Modeling of Kashmir Aftershock Decay Based on Static Coulomb Stress

  Changes and Laboratory-Derived Rate-and-State Dependent Friction Law, Pure and Applied Geophysics, 173, 1559-1574,

  doi: 10.1007/s00024-015-1192-9, 2016.
  - Jian, P. R., Hung, S. H. and Meng, L.: Rupture behavior and interaction of the 2018 Hualien earthquake sequence and its tectonic implication, Seismological Research Letters, 90 (1), 68–77. doi: https://doi.org/10.1785/0220180241, 2018.
- Jiang, Z., Huang, D., Yuan, L. et al.: Coseismic and postseismic deformation associated with the 2016 Mw 7.8 Kaikoura earthquake, New Zealand: Fault movement investigation and seismic hazard analysis. Earth, Planets and Space, 70(1), 62, https://doi.org/10.1186/s40623-018-0827-3, 2018.

- King, G. C., Stein, R. S. and Lin, J.: Static stress changes and the triggering of earthquakes, Bulletin of the Seismological Society of America, 84(3), 935–953, https://doi.org/10.1785/BSSA0840030935, 1994.
- Liao, B. Y. and Huang, H. C.: Coulomb stress changes and seismicity in central Taiwan due to the Nantou blind-thrust earthquakes in 2013, J. Asian Earth Sci., 124(1), 169-180, https://doi.org/10.1016/j. jseaes. 2016.05.001, 2016.
  - Liao, B. Y., Zhang, Y. T., Xie, S and Chi, W. H.: The source characteristics and ground motion parameters measured with structural damages of Mw6.4, 2018 Hualien Earthquake, Taiwan, from GPS data, Edelweiss Applied Science and Technology, 8, 3816–3831, https://doi.org/10.55214/25768484.v8i6.2826, 2024.
- Lin, J. and Stein, R. S.: Stress triggering in thrust and subduction earthquakes and stress interaction between the southern San Andreas and nearby thrust and strike-slip faults, Journal of Geophysical Research: Solid Earth, 109(B2), B02303, https://doi.org/10.1029/2003JB002607, 2004.
  - Molchan, G. M.: Strategies in strong earthquake prediction, Phys. Earth Planet. Inter, 61, 84–98, https://doi.org/10.1016/0031-9201(90)90097-H, 1990.
- Mignan, A, King, G. C. P.: A model for the stress transfer and release during earthquakes: Accelerating moment release revisited, Journal of Geophysical Research: Solid Earth, 112(B7), B07302, https://doi.org/10.1029/2006JB004671, 2007.
  - Nathan, P. and Walter, D. M.: Coulomb stress models for the 2019 Ridgecrest, California earthquake sequence, Tectonophysics, 791, 228555, https://doi.org/10.1016/j.tecto.2020.228555, 2020.
  - Nieh, J. H., Hsu, T. H., Cheng, H. C., Chong, K. C. and Lai, P. F.: 2018 Taiwan Hualien Earthquake—Disaster lessons we learned in the emergency department of a tertiary hospital, Journal of Acute Medicine, 10(3), 149–155, doi: 10.6705/j.jacme.202012 10(4).0003, 2020.
  - Okada, Y.: Surface deformation due to shear and tensile faults in a half-space, Bulletin of the Seismological Society of America, 75, (4), 1135–1154, doi: https://doi.org/10.1785/BSSA0750041135, 1985.
  - Pang, Y.: Stress evolution on major faults in Tien Shan and implications for seismic hazard, Journal of Geodynamics, 153–154, 101939, https://doi.org/10.1016/j.jog.2022.101939, 2022.
- Pranger, C., Sanan, P., May. D. A. et al.: Rate and State Friction as a Spatially Regularized Transient Viscous Flow Law, Journal of Geophysical Research: Solid Earth, 127(6), e2021JB023511, https://doi.org/10.1029/2021JB023511, 2022.
  - Regenauer-Lieb, K., Hu, M., Schrank, C. et al.: Cross-diffusion waves resulting from multiscale, multiphysics instabilities: application to earthquakes, Solid Earth, 12, 1829–1849, https://doi.org/10.5194/se-12-1829-2021, 2021, 2021.
- Shan, B., Xiong, X., Wang, R. et al.: Stress evolution and seismic hazard on the Maqin-Maqu segment of East Kunlun Fault zone from co-, post- and interseismic stress changes, Geophysical Journal International , 200, 244–253, https://doi.org/10.1093/gji/ggu395, 2015.
  - Shebalin, P. and Narteau, C.: Depth dependent stress revealed by aftershocks. Nat Commun, **8**, 1317. https://doi.org/10.1038/s41467-017-01446-y, 2017.

- Su, L, Shi, F., Zhai, H., Yang, C. and Wang, Y.: The earthquake probabilities of the Liupanshan fault zone based on the 900 Coulomb stress and friction constitutive law, Geomatics, Natural Hazards and Risk, 15(1). https://doi.org/10.1080/19475705.2024.2344811, 2024.
  - Tang, D., Ge, W. and Cao, X.: Stress triggering of the 2022 Lushan–Maerkang earthquake sequence by historical events and its implication for fault stress evolution in eastern Tibet, Frontiers in Earth Science, 11, 1105394, https://doi.org/10.3389/feart.2023.1105394, 2023.
- 905 Ueda, T. and Kato, A.: Aftershocks following the 2011 Tohoku-Oki earthquake driven by both stress transfer and afterslip, Progress in Earth and Planetary Science, 10, 31, https://doi.org/10.1186/s40645-023-00564-0, 2023.
  - Utsu, T. Statistical study on the occurrence of aftershocks, Geophysical Magazine, 30, 521–605, 1961.
  - Wen, Y. Y., Wen, S., Lee, Y. H. and Ching, K. E.: The kinematic source analysis for 2018 Mw 6.4 Hualien, Taiwan earthquake, Terr. Atmos. Ocean. Sci., 30, 1-11, doi: 10.3319/TAO.2018.11.15.03, 2019.
- Yiong, W., Qiao, X., Liu, G., Chen, W. and Nie, Z.: Coulomb stress evolution along the Kongur extensional system since 1895 and present seismic hazard, Geodesy and Geodynamics, 10(1), 1–9. https://doi.org/10.1016/j.geog.2018.11.007, 2019.
  Xie, C., Zhu, Y., Ji, Y. et al.: Coseismic Stress Change and Viscoelastic Relaxation after the 2008 Great Sichuan Earthquake, Applied Sciences, 12(19), 9585, https://doi.org/10.3390/app12199585, 2022.
- Yang, L., Wang, J. and Xu, C. (2024) Coseismic Coulomb stress changes induced by a 2020–2021 MW > 7.0 Alaska earthquake sequence in and around the Shumagin gap and its influence on the Alaska-Aleutian subduction interface, Geodesy and Geodynamics, 15(1), 1-12, https://doi.org/10.1016/j.geog.2023.04.007, 2024.
  - Zaccagnino, D. and Doglioni, C.: Fault dip vs. shear stress gradient, Geosystems and Geoenvironment, 2(4), 100211, https://doi.org/10.1016/j.geogeo.2023.100211, 2023.
- Zavyalov, A., Zotov, O., Guglielmi, A. and Klain, B.: On the Omori Law in the Physics of Earthquakes, Applied Sciences, 12(19), 9965, https://doi.org/10.3390/app12199965, 2022.
  - Zhang, P., Deng, Q. and Lin, S. Coulomb stress changes induced by the 2008 Wenchuan earthquake, Earthquake Science, 26(1), 1–12, https://doi.org/10.1785/0220130111, 2013.