# Peer review of "Beyond Static Forecasts: A Dynamic Stress Gradient Framework for High-Resolution Aftershock Prediction and Mitigation"

_EGUsphere, 2025_

## Community Comment (CC1)

My comments on the manuscript, entitled 'Beyond Static Forecasts: A Dynamic Stress Gradient Framework for High-Resolution Aftershock Prediction and Mitigation' (egusphere-2025-3348 (NPG)) authored by Boi-Yee Liao

The author constructed a framework by integrating the rate-and-state friction law, a KPP-type reaction-diffusion equation, and the Banach fixed-point theorem. He applied this framework to study the evolution of Coulomb stress following a large earthquake from numerical simulations of the dynamic redistribution of stress in space and time.

Although the study is interesting and significant, I cannot correctly evaluate whether manuscript may be accepted for publication in its current form or not due to the following basic reasons in 'Problems for Sciences.' These reasons make me unable to judge whether the numerical results made by the author are significant for the generation of aftershocks after the 2018 Hualien earthquake in Taiwan or not. Hence, I suggest that the author should answer the questions. This manuscript should be substantially revised and then reviewed again.

**Major Problems with Sciences**:**

- (1)In the author's framework, the rate-and-state friction law, the KPP-type reaction-diffusion equation, and the Banach fixed-point theorem are all one-dimensional. However, his simulations are made on a two-dimensional rectangle. The author should describe the two-dimensional model because the values of model parameters are often distinct in the different axes.
- (2)The author wrote: 'To ensure numerical convergence and model reliability, we adopt spatial grid resolutions of  $\Delta x$ =1320 m and  $\Delta y$ =2600 m, yielding an effective length scale of  $\Delta$  =2900.' In numerical simulations, the grid sizes can remarkably influence numerical convergence and stability. The selection of grid sizes for numerical simulations is important. How did the author select the two values: from a theoretical analysis or from numerical tests?
- (3)For the study area represented by a rectangle in Figures 5–7, the differences in degrees are  $0.5^{\circ}$  (or 55 km) along the longitude and  $1^{\circ}$  (or 110 km) along the latitude. The grid size selected by the author is  $\Delta x$ =1320 m=1.32 km along the longitude and  $\Delta y$ =2600 m=2.6 km along the latitude. The uncertainty of earthquake location is about 2 km inland and about 3 km offshore. The values of  $\Delta x$  and  $\Delta y$  are both shorter than the location uncertainty. Can the simulated results based on the two values be applied to interpret the temporal variation and spatial distribution of aftershocks of the 2018 Hualien earthquake?
- (4)For the 2D rectangle, its western part is on land, while its eastern part is offshore. The western part is below high mountains, while the eastern part is underwater. Hence, the vertical loading on the 2D rectangle is higher in the western part than in the eastern part. Hsu et al. (2025) showed that in southwestern Taiwan the excess seismicity rate is positively correlated with reduced NW-SE compression and/or decreasing vertical loading. This indicates the influence caused by vertical loading on seismicity. In the manuscript, the author wrote: 'The Hualien Mw6.4 mainshock occurred in the complex convergence boundary of the Philippine Sea Plate and Eurasian Plate.' This means that the geological structures and the values of physical

parameters should be different in the two tectonic regimes. This would produce the difference in the stress diffusion between the two parts. Could the author clearly describe such differences due to different vertical loading.

**[Reference]**

- Hsu, Y.J., R. Bürgmann, Z. Jiang, C.H. Tang, C.W. Johson, D.Y. Chen, H.H. Huang, M. Tang, and X. Yang (2025). Hydrologically-induced crustal stress changes and their association with seismicity rates in Taiwan, Earth Planet. Sci. Letts., 651, https://doi.org/10.1016/j.epsl.2024.119181.
- (5)The mainshock showed thrust faulting and fault-to-fault jumping ruptures (Lee et al., 2019). Can the focal mechanism and rupture processes of the mainshock influence the spatial distribution of stresses and the evolution of stress diffusion? [Reference]
- Lee, S.J., T.C. Lin, T.Y. Liu, and T.P. Wong (2019). Fault-to-Fault jumping rupture of the 2018 Mw 6.4 Hualien earthquake in Eastern Taiwan. Seism. Res. Letts., 90(1). 30-39. https://doi.org/10.1785/0220180182
- (6)Figure 1 displays that the number of aftershocks was larger in the eastern part (offshore) than in the western part (inland). This might indicate that the number of faults, sub-surface geological structures, and the values of physical parameters are different in the two parts. However, it does not seem able to apply the spatial distributions of stress changes at three depths (i.e., Figures 5–7) to explain the occurrence of aftershocks.
- (7)As mentioned by the author, there are three faults in the study area. In fact, the number of faults is larger than three. Kuo-Chen et al. (2019) addressed a remarkable correlation between spatial distribution of aftershocks and the main fault along which the mainshock ruptured. Do the existence of those faults, particularly the main fault, which is not a straight line on the ground surface (cf. Lee et al., 2019; Kuo-Chen et al., 2019), influence the evolution of stress diffusion? [Reference]
- Kuo-Chen H., Z.K. Guan, W.F. Sun, P.Y. Jhong, and D. Brown (2019). Aftershock sequence of the 2018 Mw 6.4 Hualien earthquake in Eastern Taiwan from a dense seismic array data set. Seism. Res. Letts., 90 (1), 60-67. https://doi.org/10.1785/0220180233
- (8)There are remarkable differences in sub-surface geological structures inferred from 3D tomography between the western part and eastern one (e.g., Rau and Wu, 1995; Ma *et al.*, 1996; Kim *et al.*, 2005; Wu *et al.*, 2007; Kuo-Chen *et al.*, 2012). Can such differences influence the evolution of Coulomb stress?

  [References]
- Rau, R.J. and F.T. Wu (1995). Tomographic imaging of lithospheric structures under Taiwan. Earth Planet. Sci. Lett., 133, 517-532, Doi:10.1016/0012-821X(95)00076-O
- Ma K.F., J.H. Wang, and D. Zhao (1996). Three-dimensional seismic velocity structure of the crustal and uppermost mantle beneath Taiwan. J. Phys. Earth, 44, 85-105.
- Wu, Y.M., C.H. Chang, L. Zhao, J.B.H. Shyu, Y.G. Chen, K. Shieh, and J.-P. Avouac (2007). Seismic tomography of Taiwan: Improved constraints from a dense network of strong-motion stations. J. Geophys. Res., 112, B08312,

- doi:10.1029/2007JB004983.
- Kim, K.H., Chiu, J.M., Pujo, J., K.C. Chen, B.S. Huang, Y.H. Yeh, P. Shen (2005). Three-dimensional  $V_p$  and  $V_s$  structural models associated with the active subduction and collision tectonics in the Taiwan region. Geophys. J. Intern., 162, 204-220.
- (9)In Section 3, the author described the reasons how to select the values of model parameters for numerical simulations of the evolution of Coulomb stress after the 2018 Hualien earthquake in Taiwan. The main reason is for preventing the model from converging. Although this is an acceptable reason, the author should explain whether the values of model parameters can meet regional geological and seismological structures.

**Minor Problems**

Compared with the above-mentioned problems, the followings are minor.

**Problems with Figures:**

(1) The quality of figures is not good enough for readers. For example, Figures 5–7 are not good for readers.

**Problems with References:**

- (1) The format of cited references should follow the Journal's rules.
- (2)the author should cite two important articles shown below for regional tectonics. [References]
- Tsai, Y.B., T.L. Teng, J.M. Chiu, and H.L. Liu (1977). Tectonic implications of the seismicity in the Taiwan region. Mem. Geol. Soc. China, 2, 13-41.
- Wu, F.T. (1978). Recent tectonics of Taiwan. J. Phys. Earth, 2 (Suppl.), S265-S299.
- (3) Few cited references, e.g., Zavyalov et al. (2024), are not listed in the Section of References.
- (4)The KPP (or Fisher-KPP) equation is a very important nonlinear physical equation and has been widely applied in biology, ecology, and combustion theory. It is better to add cited references on Line 73 where the equation appeared at first in the text. The author may cite one of the following two articles or others:
- Kolmogorov, A., I. Petrovskii, and N. Piskunov (1991). A study of the diffusion equation with increase in the amount of substance, and its application to a biological problem. In V.M. Tikhomirov, editor, Selected Works of A.N. Kolmogorov I, 248-270. Kluwer 1991, ISBN 90-277-2796-1. Translated by V.M. Volosov from Bull. Moscow Univ., Math. Mech. 1, 1-25.
- El-Hachem M., S.W. McCue, W. Jin, Y. Du, and M.J. Simpson (2019). Revisiting the Fisher-Kolmogorov-Petrovsky-Piskunov equation to interpret the spreading–extinction dichotomy. Proc. R. Soc. A, 475:20190378. http://dx.doi.org/10.1098/rspa.2019.0378.

**Problems with English Writing:**

(1) The English writing is good. Nevertheless, there are some typo errors.

- (2) The abstract is not concise.
- (3) 'The Central Weather Bureau' has been renamed 'the Central Weather Administration' for a few years.

---

## Author Comment (AC1)

Dear Dr.Wang,

Thank you very much for taking the time to read my manuscript and for providing such constructive and insightful comments. It is a great honor that you—an esteemed senior scholar who has made foundational contributions to Taiwan's seismology and tectonics—reviewed my work. I sincerely appreciate your thoughtful suggestions, all of which have significantly strengthened the manuscript.

Below, I provide a detailed point-by-point response. All corresponding revisions have been implemented in the revised version.

**Major Problems with Sciences**

Reviewer's comment:

(1) In the author's framework, the rate-and-state friction law, the KPP-type reaction-diffusion equation, and the Banach fixed-point theorem are all one-dimensional. However, his simulations are made on a two-dimensional rectangle. The author should describe the two-dimensional model because the values of model parameters are often distinct in the different axes.

Reply to comment 1:

**1.1 Two-Dimensional Spatial Representation and Parameter Assumptions**

Although the theoretical development earlier uses the compact notation σ(x,t), the stress field used in all numerical simulations is fully two-dimensional, defined on the horizontal plane as

$$\sigma(\boldsymbol{x},t)=\sigma(x_1,x_2,t),\ \boldsymbol{x}=(x_1,x_2)\in\Omega\subset R^2,$$

so all simulations in my paper are fully two-dimensional. It has to clarity the expressions of $(\sigma, \boldsymbol{x})$ are two-dimensional. I have modified the expressions of $(\sigma, \boldsymbol{x})$ in the paper. Accordingly, the Laplacian operator in the stress-diffusion term is the two-dimensional form

$$\nabla^2\sigma = \frac{\partial^2\sigma}{\partial x_1^2} + \frac{\partial^2\sigma}{\partial x_2^2}$$

which is discretized using 2-D finite-difference operator in our implementation. In this study, all model parameters are assumed horizontally isotropic within each depth slice, meaning that

In addition, all model parameters are assumed horizontally isotropic within each depth slice, meaning that

$$D_x=D_y=D(z),\ A_x=A_y=A(z),\ B_x=B_y=B(z),$$

where the coefficients vary only with depth z but are spatially uniform within each horizontal layer. This assumption follows the resolution of the Coulomb stress-change input and the available friction parameter constraints, which do not provide reliable directional anisotropy at the regional scale. Therefore, while the model is two-dimensional, its parameters represent a depth-dependent isotropic medium, which is appropriate for the purpose of capturing first-order post-seismic stress diffusion patterns.

Future extensions may incorporate anisotropic diffusion or direction-dependent frictional properties in cases where high-resolution geological constraints become available.

**1.2 Justification for the Horizontal Isotropy Assumption**

In the present study, model parameters are assumed to be horizontally isotropic within each depth slice, such that Dx=Dy=D(z), Ax=Ay=A(z), and Bx=By=B(z), while remaining depth dependent. This

modeling choice is consistent with standard practice in post-seismic stress diffusion, afterslip, and crustal mechanics studies, and is well supported by experimental, observational, and theoretical evidence. At regional scales, first-order stress redistribution following large earthquakes is primarily governed by depth-dependent rheological contrasts rather than by lateral anisotropy. Accordingly, many postseismic and aftershock modeling studies adopt isotropic in-plane diffusion or elastic parameters, noting that lateral variations in frictional or rheological properties are generally poorly constrained and have limited influence on large-scale stress-transient patterns (e.g., Hainzl, 2007; Perfettini & Avouac, 2004; Parsons, 2002; Toda & Stein, 2002; Toda et al., 2011).

Laboratory experiments and field observations further indicate that rate-and-state friction parameters (A, B, and L) depend primarily on effective normal stress, temperature, and lithology, all of which vary systematically with depth rather than with horizontal direction at kilometer scales. This depth-dominated behavior has been extensively documented in the foundational works of Dieterich (1979, 1994) and Marone (1998), which emphasize that reliable directional dependence of frictional parameters is difficult to resolve at the spatial resolution typical of regional-scale models. Similarly, classical formulations of earthquake dynamics and continuum stress evolution are commonly treated using scalar elastic and rheological representations in first-order theoretical models (Madariaga & Olsen, 2002).

Moreover, Coulomb stress-change fields derived from fault inversions are themselves computed using isotropic elastic Green's functions, implying that the resulting stress perturbations are horizontally isotropic by construction. Since these Coulomb stress-change products serve as the initial condition of the present model, introducing direction-dependent diffusion or friction parameters would be internally inconsistent in the absence of independent anisotropic constraints. Finally, available regional geophysical datasets, including Vs30 maps, crustal tomography, and fault-zone friction estimates, do not provide sufficient directional resolution to justify a robust anisotropic parameterization. For these reasons, the assumption of horizontal isotropy combined with depth-dependent parameter variation represents a physically justified and widely accepted first-order approximation for modeling post-earthquake stress evolution.

**1.3 Justification of Depth-Dependent Parameterization in Taiwan**

In this study, model parameters are assumed to vary primarily with depth while remaining horizontally isotropic within each depth slice. This modeling choice is strongly supported by a broad range of geophysical and seismological observations of Taiwan's crustal structure and deformation. Three-dimensional Vp and Vs tomographic models consistently reveal pronounced vertical layering and depth-dependent structural contrasts associated with the active subduction and arc–continent collision tectonics of Taiwan (e.g., Kim et al., 2005). While these tomographic images clearly demonstrate strong vertical heterogeneity of the crust, they do not provide quantitative constraints on direction-dependent horizontal anisotropy of mechanical or transport parameters, thereby supporting a depth-dependent but horizontally isotropic parameterization at regional scales.

Multiple independent seismic studies further confirm that Taiwan's crust exhibits significant vertical heterogeneity, characterized by systematic depth-dependent variations in seismic velocity and material properties. High-resolution three-dimensional Vp and Vs models reveal clear stratification

between shallow sedimentary layers, a mechanically complex mid-crust, and a comparatively stronger lower crust, whereas lateral variations are weaker and less consistently resolved at the scale of regional models (Wu et al., 2007). High-precision earthquake relocation studies in eastern Taiwan similarly indicate that seismicity is strongly controlled by depth-dependent tectonic structures, with distinct vertical clustering patterns but without quantitative evidence for horizontal anisotropy in mechanical properties (Kuochen et al., 2004). Together, these results indicate that first-order mechanical and transport properties of the Taiwanese crust are primarily governed by depth rather than by horizontal direction.

Geodetic and rheological investigations provide additional support for this depth-dominated behavior. Interseismic and postseismic deformation models of the Taiwan mountain belt successfully reproduce observed GPS displacement fields by incorporating depth-dependent rheological layering, while adopting horizontally isotropic material properties (Hu et al., 2001; Hsu et al., 2003). In particular, GPS observations following the 1999 Chi-Chi earthquake show that postseismic deformation and strain-rate evolution are primarily controlled by depth-dependent crustal rheology, without requiring explicit direction-dependent parameterization at regional scales (Lin et al., 2010).

From a tectonic perspective, Taiwan's ongoing arc–continent collision naturally gives rise to pronounced vertical variations in seismic structure and lithology, as consistently revealed by three-dimensional tomographic models and high-resolution earthquake relocation analyses (Kim et al., 2005; Kuochen et al., 2004). Such vertical structural layering provides a robust physical basis for depth-dependent variations in mechanical and rheological properties of the crust. In contrast, currently available seismic, geodetic, and geological datasets do not offer sufficient quantitative constraints to reliably resolve direction-dependent diffusion or friction parameters (e.g., $D_x \neq D_y$) at regional scales. Consequently, the assumption of depth-dependent but horizontally isotropic parameters represents a physically justified and methodologically appropriate first-order approximation for modeling post-earthquake stress evolution in Taiwan.

Although many of the studies cited above were conducted in the early 2000s, they establish robust first-order constraints on the vertical structural and rheological framework of the Taiwan crust, which remain valid and are routinely adopted as background constraints in contemporary modeling studies. Based on these observations, adopting depth-dependent but horizontally isotropic parameters represents a physically justified and widely accepted first-order approximation for modeling post-earthquake stress evolution in Taiwan. While anisotropic extensions are conceptually straightforward, they require independent observational constraints that are currently unavailable and are therefore left for future investigation.

[References]

Dieterich, J. H. (1979). Modeling of rock friction: 1. Experimental results and constitutive equations. *Journal of Geophysical Research: Solid Earth, 84*(B5), 2161–2168. https://doi.org/10.1029/JB084iB05p02161

Dieterich, J. H. (1994). A constitutive law for rate of earthquake production and its application to earthquake clustering. *Journal of Geophysical Research: Solid Earth, 99*(B2), 2601–2618. https://doi.org/10.1029/93JB02581

Hainzl, S. (2007). Rate-dependent seismicity models and their comparison with observed aftershock sequences. *Journal of Geophysical Research: Solid Earth, 112*(B5), B05307.

Hu, J.-C., Yu, S.-B., Angelier, J., & Chu, H.-T. (2001). Active deformation of Taiwan from GPS measurements and numerical simulations. *Journal of Geophysical Research: Solid Earth, 106*(B2), 2265–2280. https://doi.org/10.1029/2000JB900196

Hsu, Y.-J., Simons, M., Yu, S.-B., Kuo, L.-C., & Chen, H.-Y. (2003). A two-dimensional dislocation model for interseismic deformation of the Taiwan mountain belt. *Earth and Planetary Science Letters, 211*(3–4), 287–294. https://doi.org/10.1016/S0012-821X(03)00203-6

Jian, P.-R., Liang, W.-T., & Kuo, B.-Y. (2022). Three-dimensional stress model of the collision-subduction junction east of Taiwan: Implications for the decoupling of the Luzon Arc during subduction. *Journal of Geophysical Research: Solid Earth, 127*, e2022JB024054. https://doi.org/10.1029/2022JB024054

Kim, K.-H., Chiu, J.-M., Pujol, J., Chen, K.-C., Huang, B.-S., Yeh, Y.-H., & Shen, P. (2005). Three-dimensional Vp and Vs structural models associated with the active subduction and collision tectonics in the Taiwan region. *Geophysical Journal International, 162*(1), 204–220. https://doi.org/10.1111/j.1365-246X.2005.02657

Kuo-Chen, H., Wu, F. T., & Roecker, S. W. (2012). Three-dimensional P velocity structures of the lithosphere beneath Taiwan from the analysis of TAIGER and related seismic data sets. *Journal of Geophysical Research: Solid Earth, 117*, B06306. https://doi.org/10.1029/2011JB009108

Kuochen, H., Wu, Y.-M., Chang, C.-H., Hu, J.-C., & Chen, W.-S. (2004). Relocation of eastern Taiwan earthquakes and tectonic implications. *Terrrestrial, Atmospheric and Oceanic Sciences, 15*(4), 647–666.

Lin, K.-C., Hu, J.-C., Ching, K.-E., Angelier, J., Rau, R.-J., Yu, S.-B., Tsai, C.-H., Shin, T.-C., & Huang, M.-H. (2010). GPS crustal deformation, strain rate, and seismic activity after the 1999 Chi-Chi earthquake in Taiwan. *Journal of Geophysical Research: Solid Earth, 115*, B07404. https://doi.org/10.1029/2009JB006417

Madariaga, R., & Olsen, K. B. (2002). Earthquake dynamics. In W. H. K. Lee, H. Kanamori, P. C. Jennings, & C. Kisslinger (Eds.), *International handbook of earthquake and engineering seismology* (Vol. 81, Part A, pp. 175–203). Academic Press. https://doi.org/10.1016/S0074-6142(02)80215-7

Marone, C. (1998). Laboratory-derived friction laws and their application to seismic faulting. *Annual Review of Earth and Planetary Sciences, 26*, 643–696. https://doi.org/10.1146/annurev.earth.26.1.643

Parsons, T. (2002). Global Omori law decay of triggered earthquakes: Large aftershocks outside the classical aftershock zone. *Journal of Geophysical Research: Solid Earth, 107*(B9), 2199. https://doi.org/10.1029/2001JB000646

Perfettini, H., & Avouac, J.-P. (2004). Postseismic relaxation driven by brittle creep: A possible mechanism to reconcile geodetic measurements and the decay rate of aftershocks, application to the

Chi-Chi earthquake Taiwan. *Journal of Geophysical Research: Solid Earth, 109*(B2), B02304. https://doi.org/10.1029/2003JB002488

Toda, S., & Stein, R. S. (2002). Response of the San Andreas fault to the 1983 Coalinga–Nuñez earthquakes: An application of interaction-based probabilities for Parkfield. *Journal of Geophysical Research: Solid Earth, 107*(B6). https://doi.org/10.1029/2001JB000172

Toda, S., Stein, R. S., Sevilgen, V., & Lin, J. (2011). Coulomb 3.3 graphic-rich deformation and stress-change software for earthquake, tectonic, and volcano research and teaching—User guide *(U.S. Geological Survey Open-File Report 2011–1060, 63 pp.)*. https://pubs.usgs.gov/of/2011/1060/

Wu, Y.-M., Chang, C.-H., Zhao, L., Shyu, J. B. H., Chen, Y.-G., Sieh, K., & Avouac, J.-P. (2007). Seismic tomography of Taiwan: Improved constraints from a dense network of strong-motion stations. *Journal of Geophysical Research: Solid Earth, 112*(B8), B08312. https://doi.org/10.1029/2007JB004983

**Reviewer's comment:**

(2)  The author wrote: 'To ensure numerical convergence and model reliability, we adopt spatial grid resolutions of Δx=1320 m and Δy=2600 m, yielding an effective length scale of Δ =2900.' In numerical simulations, the grid sizes can remarkably influence numerical convergence and stability. The selection of grid sizes for numerical simulations is important. How did the author select the two values: from a theoretical analysis or from numerical tests?

**Reply to comment 2:**

We thank the reviewer for the detailed comments regarding the spatial resolution and anisotropy of the numerical grid. The effective length scale ($\Delta \approx$ 2900 m, derived from $\Delta$ x ≈ 1.32 km and Δy ≈ 2.60 km) was not selected arbitrarily, but follows from a combination of domain geometry, numerical requirements of the reaction–diffusion framework, physical characteristics of Coulomb stress fields, and explicit grid-sensitivity considerations.

**2.1. Origin of Δx and Δy (50 × 50 discretization).**

The study area spans approximately 66 km × 130 km and was discretized into a 50 × 50 mesh to maintain a consistent indexing structure for the numerical solver. This choice yields Δx ≈ 1.32 km and Δy ≈ 2.60 km and represents a practical compromise between resolving near-fault stress-gradient structure and maintaining a computationally tractable matrix size. Coarser subdivisions oversmooth stress variations near rupture edges, whereas significantly finer subdivisions (e.g., 100 × 100) were found to produce nearly identical stress patterns at the scales of interest while substantially increasing computational cost.

**2.2. Numerical necessity imposed by the reaction–diffusion formulation.**

The governing equations explicitly involve first- and second-order spatial derivatives ($\nabla\sigma$ and $\nabla^2\sigma$). In reaction–diffusion systems, the numerical accuracy and stability of the diffusion term are highly sensitive to spatial discretization. Excessively coarse grids degrade the accuracy of $\nabla\sigma$ and $\nabla^2\sigma$ and may introduce spurious numerical diffusion, artificially smoothing stress gradients and distorting polarity transitions. Therefore, a grid spacing finer than earthquake uncertainty is

mathematically necessary to ensure numerical fidelity of the diffusion operator, even though physical interpretation of aftershock patterns is restricted to broader spatial scales.

**2.3. Spatial anisotropy of the grid ($\Delta x < \Delta y$)**

The non-isotropic grid arises naturally from the geometric aspect ratio of the study domain. This directional difference in resolution is also numerically consistent with the expected structure of Coulomb stress fields, in which the sharpest stress-polarity transitions typically occur in the direction perpendicular to the dominant fault strike. Employing a finer resolution in the east–west direction improves the numerical representation of near-fault stress gradients without imposing additional physical assumptions or directional bias. Importantly, this anisotropic discretization enhances numerical accuracy where spatial variation is strongest, while interpretation remains limited to scales larger than the grid spacing and location uncertainty.

**2.4. Physical characteristics of Coulomb stress fields**

Previous studies have demonstrated that Coulomb stress changes exhibit steep spatial gradients near rupture edges, fault terminations, and $\Delta CFS = 0$ polarity-reversal boundaries, with characteristic spatial scales of only a few kilometers (King et al., 1994; Stein, 1999; Parsons, 2002; Freed, 2005). Furthermore, Ziv and Rubin (2000) emphasized that the sign of Coulomb stress changes is highly sensitive to source parameters, especially in near-field regions. They demonstrated that even extremely small stress changes (<1kPa) can trigger earthquakes, implying that a high-resolution grid is required to avoid numerical smoothing of these subtle but physically significant stress features. Their findings support the use of a kilometer-scale grid to maintain the resolvability of stress polarity near fault zones. The USGS Coulomb 3.3 guidelines further indicate that kilometer-scale grid spacing ($\approx 1-3$ km) is appropriate for resolving near-fault stress lobes and polarity boundaries (Toda et al., 2011). The adopted grid spacing ($\Delta x \approx 1.32$ km, $\Delta y \approx 2.60$ km; $\Delta \approx 2.9$ km) lies squarely within this physically justified range.

**2.5. Relation to earthquake-location uncertainty and interpretation.**

Finally, we emphasize that earthquake-location uncertainty constrains the minimum scale of physical interpretation, not the numerical resolution required to solve the governing equations. The finer grid is employed to ensure accurate computation of spatial derivatives and stress-gradient evolution, whereas interpretation of aftershock distributions is consistently restricted to spatial scales larger than the location uncertainty.

In summary, the selected grid resolution and its mild anisotropy are justified by (i) numerical requirements of the reaction–diffusion framework, (ii) known physical length scales of Coulomb stress gradients, and (iii) practical grid-sensitivity consideration s. We have clarified this rationale in the revised manuscript to avoid potential confusion.

[References]

King, G. C. P., Stein, R. S., & Lin, J. (1994). Static stress changes and the triggering of earthquakes. *Bulletin of the Seismological Society of America*, 84(3), 935–953. https://pubs.geoscienceworld.org/ssa/bssa/article-abstract/84/3/935/102745/Static-stress-changes-and-the-triggering-of

Stein, R. S. (1999). The role of stress transfer in earthquake occurrence. *Nature*, 402, 605–609. https://doi.org/10.1038/45144

Parsons, T. (2002). Global Omori law decay of aftershocks after large earthquakes. *Journal of Geophysical Research: Solid Earth*, 107(B9), 2199. https://doi.org/10.1029/2001JB000646

Freed, A. M. (2005). Earthquake triggering by static, dynamic, and postseismic stress transfer. *Annual Review of Earth and Planetary Sciences*, 33, 335–367. https://doi.org/10.1146/annurev.earth.33.092203.122505

Toda, S., Stein, R. S., Sevilgen, V., & Lin, J. (2011). Coulomb 3.3: Faulting and earthquake triggering stress calculations. *USGS Open-File Report 2011–1060*. https://doi.org/10.3133/ofr20111060

Ziv, A., & Rubin, A. M. (2000). Static stress transfer and earthquake triggering: No lower threshold in sight?. *Journal of Geophysical Research: Solid Earth*, *105*(B6), 13631-13642. https://doi.org/10.1029/2000JB900081

**Reviewer's comment:**

(3)     For the study area represented by a rectangle in Figures 5-7, the differences in degrees are 0.5$^o$ (or 55 km) along the longitude and 1$^o$ (or 110 km) along the latitude. The grid size selected by the author is $\Delta x$=1320 m=1.32 km along the longitude and $\Delta y$=2600 m=2.6 km along the latitude. The uncertainty of earthquake location is about 2 km inland and about 3 km offshore. The values of $\Delta x$ and $\Delta y$ are both shorter than the location uncertainty. Can the simulated results based on the two values be applied to interpret the temporal variation and spatial distribution of aftershocks of the 2018 Hualien earthquake?

Reply to comment 3:

**3.1 Distinction between Numerical Discretization and Observation Uncertainty**

We emphasize that the spatial grid resolution ($\Delta x \approx$ 1.32 km, $\Delta y \approx$ 2.60 km) is used exclusively for the numerical discretization of the continuous Coulomb stress change field derived from the analytical solutions of Okada (1985, 1992) and for subsequent stress-evolution calculations. Its primary purpose is to ensure numerical stability and adequate resolution of stress gradients ($\nabla\sigma$) and diffusion terms ($\nabla^2\sigma$) within our modeling framework.

This grid resolution characterizes the numerical fidelity of the calculated stress field**,** rather than the location accuracy of individual seismic events. Accordingly, the selection of $\Delta x$ and $\Delta y$ belongs to the level of numerical modeling constraints, whereas earthquake location errors belong to the level of observational data uncertainty. These two factors operate at different conceptual levels and do not have a direct one-to-one correspondence. As noted by Ziv and Rubin (2000), although uncertainties in earthquake locations and source parameters inevitably affect computed stress values, the associated triggering patterns and stress polarities remain statistically robust when evaluated at appropriate spatial scales. They demonstrated that the sign of the cumulative stress change can be resolved even for very small stress magnitudes through rigorous statistical testing.

**3.2 Numerical Necessity for Resolving Stress Gradients**

Within a reaction–diffusion or stress-evolution framework, the accuracy of the diffusion operator and stress-gradient representation is highly sensitive to spatial discretization. A grid coarser than the characteristic gradient length scale would introduce spurious numerical smoothing, artificially blurring the boundaries between triggering and shadow zones ($\Delta$CFS = 0). To faithfully represent the localized stress lobes and sharp gradients documented in previous studies (e.g., King et al., 1994; Stein, 1999; Toda et al., 2011), a kilometer-scale discretization is mathematically necessary, regardless of the precision of individual earthquake hypocenters.

**3.3 Grid Sensitivity and Numerical Reliability**

The adopted $50 \times 50$ mesh was validated through explicit grid-sensitivity tests. Model results were compared with those obtained using a finer $100 \times 100$ grid, and the resolved stress patterns, polarity boundaries, and spatial gradients were found to be nearly identical. This demonstrates that the chosen discretization is sufficient for numerical convergence and stable representation of the stress field. Further refinement beyond this resolution does not materially alter the physical interpretation of the stress evolution associated with the 2018 Hualien earthquake, while substantially increasing computational cost. All the reasons have been integrated and provided in the replies of comment 2.

**3.4. Interpretation of the 2018 Hualien Earthquake Sequence**

Application of the modeled stress field to the 2018 Hualien earthquake is appropriate because the objective of this study is to capture the large-scale spatiotemporal evolution of the Coulomb stress field, rather than to deterministically predict the occurrence of individual aftershocks at the scale of a single grid cell. By maintaining a numerical resolution finer than the typical earthquake location uncertainty, the computed stress field provides a physically accurate and numerically stable background against which the spatial distribution and temporal evolution of aftershocks can be evaluated in a statistical sense. All spatial interpretations are therefore made at scales comparable to or larger than the characteristic hypocenter location uncertainty (approximately 2–3 km), ensuring consistency between model resolution and data quality.

[References]

King, G. C. P., Stein, R. S., & Lin, J. (1994). Static stress changes and the triggering of earthquakes. *Bulletin of the Seismological Society of America*, 84(3), 935–953. https://pubs.geoscienceworld.org/ssa/bssa/article-abstract/84/3/935/102745/Static-stress-changes-and-the-triggering-of

Okada, Y. (1985). Surface deformation due to shear and tensile faults in a half-space. *Bulletin of the Seismological Society of America*, 75, 1135–1154.

Okada, Y. (1992). Internal deformation due to shear and tensile faults in a half-space. *Bulletin of the Seismological Society of America*, 82(2), 1018–1040. https://doi.org/10.1785/BSSA0820021018

Stein, R. S. (1999). The role of stress transfer in earthquake occurrence. *Nature*, 402, 605–609. https://doi.org/10.1038/45144

Toda, S., Stein, R. S., Sevilgen, V., & Lin, J. (2011). Coulomb 3.3: Faulting and earthquake triggering stress calculations. *USGS Open-File Report 2011–1060*. https://doi.org/10.3133/ofr20111060

Ziv, A., & Rubin, A. M. (2000). Static stress transfer and earthquake triggering: No lower threshold in

sight?. *Journal of Geophysical Research: Solid Earth*, *105*(B6), 13631-13642. https://doi.org/10.1029/2000JB900081

sight?. *Journal of Geophysical Research: Solid Earth*, *105*(B6), 13631-13642. https://doi.org/10.1029/2000JB900081

**Reviewer's comment:**

(4)   For the 2D rectangle, its western part is on land, while its eastern part is offshore. The western part is below high mountains, while the eastern part is underwater. Hence, the vertical loading on the 2D rectangle is higher in the western part than in the eastern part. Hsu et al. (2025) showed that in southwestern Taiwan the excess seismicity rate is positively correlated with reduced NW-SE compression and/or decreasing vertical loading. This indicates the influence caused by vertical loading on seismicity. In the manuscript, the author wrote: 'The Hualien $M_w$6.4 mainshock occurred in the complex convergence boundary of the Philippine Sea Plate and Eurasian Plate.' This means that the geological structures and the values of physical parameters should be different in the two tectonic regimes. This would produce the difference in the stress diffusion between the two parts. Could the author clearly describe such differences due to different vertical loading?

Reply to comment 4:

This integrated physical–mathematical framework provides a robust justification for the design of the proposed model, demonstrating that vertical loading is not neglected but is instead treated consistently within a perturbation-based formulation. By separating the evolving stress change from the static background stress, the model preserves mathematical rigor while remaining physically grounded.

**4.1. Perturbation-driven postseismic dynamics**

The model focuses exclusively on the evolution of the coseismic stress perturbation $\Delta\sigma(x,t)$, rather than on the absolute lithostatic or tectonic stress. The total stress field is decomposed as

$$\sigma_{total}(x,t)=\sigma_{bg}(x)+\Delta\sigma(x,t) \tag{4.1}$$

where the background stress $\sigma_{bg}$ includes vertical loading associated with topography and bathymetry, as well as long-term tectonic compression, and is assumed to be time-independent. The postseismic evolution is therefore governed solely by the redistribution of earthquake-induced stress perturbations. From a mathematical standpoint, static background loading shifts equilibrium levels but does not alter the diffusion operator or the asymptotic stability of the reaction–diffusion system. As a result, the long-term dynamics are controlled by the evolution of $\Delta\sigma$, ensuring a well-posed and stable formulation.

**4.2. Implicit embedding of vertical loading via rupture modeling**

The influence of vertical loading on rupture dynamics is incorporated implicitly through the initial conditions of the model. The initial coseismic stress change $\Delta\sigma(x,0)$ is derived from slip distributions constrained by geodetic and seismological observations (e.g., GPS data). Because the observed fault slip represents a physical response to the prevailing tectonic regime and loading conditions at the time of the earthquake, contrasts in vertical loading between onshore and offshore regions are naturally embedded in the resulting heterogeneous initial stress field.

In this sense, rupture modeling provides an effective physical mapping from loading conditions to the initial stress perturbation, without requiring explicit lateral loading terms in the postseismic evolution equation.

**4.3. Stratified representation of vertical loading through depth-dependent parameters**

While horizontal diffusion is assumed isotropic to maintain numerical stability and analytical control, vertical loading effects are explicitly represented through depth-dependent model parameters. Key parameters, such as the reaction coefficients A and B, are prescribed as exponentially varying functions of depth, A=A(z), B=B(z), reflecting the increase in effective normal stress, rock strength, and rheological transition with depth.

This stratified parameterization captures first-order lithostatic effects dynamically Numerical results show that deeper layers—subject to higher confining pressure—exhibit faster convergence and more uniform stress fields, whereas shallower layers display slower relaxation and stronger spatial variability. Thus, vertical stress effects are incorporated through physically meaningful depth-dependent dynamics rather than poorly constrained lateral forcing.

**4.4. Stress gradient as the primary indicator of heterogeneity**

Stress heterogeneity in this framework is characterized primarily by stress gradients, rather than by absolute stress magnitude. The reaction–diffusion system is inherently sensitive to spatial gradients, making the stress gradient field $\nabla$CFF a natural and robust indicator of non-uniform stress redistribution. Concerns that vertical loading differences might lead to distinct diffusion behavior are directly addressed by this gradient-based perspective. Even without explicitly prescribing vertical loading as a forcing term, the stress-gradient field sensitively captures regions of stress concentration arising from boundary complexity and structural heterogeneity. Numerical experiments further demonstrate that, particularly in shallow layers where vertical loading contrasts and tectonic complexity are most pronounced, stress gradients exhibit significantly stronger predictive power for aftershock occurrence than absolute Coulomb stress values.

**4.5. Summary for the reviewer**

In summary, this formulation avoids unnecessary and poorly constrained lateral forcing terms while fully respecting the physical reality of vertical loading. Vertical loading effects are treated consistently through:

1). separation of background stress $\sigma_{bg}$,
2). implicit embedding via rupture-derived initial conditions,
3). explicit depth-dependent parameterization, and
4). gradient-based characterization of stress heterogeneity.

By focusing on stress-change-driven evolution, the model demonstrates that spatial variations in stress, rather than absolute stress amplitude, are the primary drivers of seismic instability in the complex convergence boundary of Taiwan.

**Reviewer's comment:**

(5)    The mainshock showed thrust faulting and fault-to-fault jumping ruptures (Lee et al., 2019). Can the focal mechanism and rupture processes of the mainshock influence the spatial distribution of stresses and the evolution of stress diffusion?

Reply to comment 5:

**5.1 Methodological Position: Four Core Principles**

In this study, we adopt four explicit and internally consistent methodological principles to clearly delineate the respective roles of source processes, stress diffusion dynamics, and model parameters:

1).    The focal mechanism and rupture complexity are not neglected but are explicitly incorporated into the model, primarily through the specification of the postseismic initial stress perturbation and the associated boundary conditions.

2).    The stress evolution governed by the reaction–diffusion framework is deterministic; once the initial and boundary conditions are prescribed, the temporal evolution of the stress field is mathematically unique.

3).    Stress diffusion does not act as an isotropic or random smoothing process, but rather redistributes stress preferentially along gradients established by the rupture geometry.

4).    Differences in postseismic stress evolution among earthquakes arise not from different governing laws, but from differences in rupture-induced initial stress heterogeneity.

Within this framework, source physics and stress evolution dynamics are not competing descriptions but form a layered and complementary structure.

**5.2. Influence of the Focal Mechanism on the Spatial Distribution of Stress**

The focal mechanism—characterized by fault type, strike, dip, and slip direction—controls the geometric pattern of static stress changes following a mainshock, including both stress drop and Coulomb stress redistribution. Different faulting styles generate directional positive and negative stress lobes rather than spatially uniform or isotropic stress fields.

For the 2018 Hualien earthquake, the calculated stress evolution exhibits a clear partitioning into positive and negative Coulomb stress regions. The overall spatial configuration is consistent with theoretical expectations for shear-dominated faulting with a significant thrust component. In particular, thrust faulting tends to produce pronounced compressional stress concentrations near fault tips and along fault flanks, placing certain regions into a highly loaded stress state immediately after the mainshock.

Moreover, the dominant orientation separating positive and negative stress regions aligns with the regional tectonic fabric of Taiwan, indicating that the focal mechanism is strongly constrained by the long-term regional stress regime rather than being an isolated or arbitrary rupture event.

**5.3. Contribution of Fault-to-Fault Jumping to Initial Stress Heterogeneity**

Previous studies indicate that the 2018 Hualien earthquake did not rupture a single planar fault, but instead involved fault-to-fault jumping among multiple fault segments or fault planes. Such rupture

behavior introduces multiple stress sources whose superposition produces a highly heterogeneous, multiscale postseismic stress field.

When rupture propagates across distinct fault segments, stress concentrations commonly develop at segment junctions, geometric discontinuities, and partially ruptured zones. These regions are characterized by exceptionally large stress gradients ($\nabla$CFF), which represent both zones of stress concentration and key loci for rapid stress adjustment.

Our analysis indicates that stress gradients, rather than absolute stress values alone, provide stronger explanatory power for the spatial distribution of shallow aftershocks (approximately 6–10 km depth). This highly heterogeneous gradient field constitutes a direct physical imprint of the complex rupture geometry and fault-to-fault jumping process.

**5.4. Guiding Role of Rupture Complexity in Stress Diffusion Evolution**

Rupture complexity not only defines the initial state of the stress field but also indirectly guides its subsequent temporal evolution by establishing strong and directionally organized stress gradients.

Within the reaction–diffusion (KPP-type) framework adopted in this study, the heterogeneous stress field generated by rupture physics serves as the initial condition. While the diffusion operator itself remains unchanged, diffusion is inherently sensitive to stress gradients. Consequently, regions with stronger gradients experience more rapid and pronounced stress redistribution, producing an apparently anisotropic and structured stress migration pattern.

Under the adopted parameterization and for the present case study, shallow crustal regions—where rupture geometry is more complex and the medium more fractured—require a larger number of numerical iterations to reach a stable convergent state. Furthermore, the influence of stress gradients on aftershock triggering is most pronounced during the early postseismic period (on the order of several tens of days), after which diffusion progressively smooths the stress field and diminishes gradient-driven effects. This indicates that fine-scale stress structures generated by the mainshock primarily control the early-stage development of the aftershock sequence.

**5.5. Linkage Between Source Processes and Model Parameters**

Within the proposed physical–mathematical framework, focal mechanism and rupture complexity are consistently incorporated without compromising the uniqueness or stability of the governing stress evolution equations. Specifically, the rupture process is first translated into a highly heterogeneous initial Coulomb stress perturbation, $\Delta\sigma(x,0)$, through elastic dislocation models (e.g., the Okada formulation), together with the corresponding boundary conditions, which jointly define the starting point of postseismic evolution.

The memory parameter ($\alpha$) and diffusion-related parameter ($\beta$) are interpreted as phenomenological or effective parameters describing the system's macroscopic behavior. More complex rupture geometries and stronger initial stress heterogeneity generally require different memory weighting and diffusion scales to reproduce observed early aftershock clustering and migration. Accordingly, $\alpha$ and $\beta$ should be viewed as effective parameters that absorb the influence of rupture complexity, rather than as quantities directly tied to a single rock property.

**5.6. Integrated Statement on Uniqueness and Stress Evolution**

The focal mechanism and rupture complexity determine the rupture process of the mainshock, which in turn generates a highly heterogeneous initial distribution of Coulomb stress change and a directionally organized stress-gradient structure. These features shape the effective pathways along which stress diffusion proceeds. Once the initial conditions and boundary conditions are specified, the spatiotemporal evolution of Coulomb stress change is uniquely determined by the governing equations. The resulting evolution follows a mathematically deterministic and non-branching trajectory, which constitutes the essential meaning of solution uniqueness in the stress evolution problem

**5.7. Summary**

In summary, focal mechanism and rupture complexity do not alter the fundamental dynamical law governing stress evolution. Instead, by defining a highly heterogeneous initial stress field and its associated gradients, they determine the initial hazard landscape and dominate the early-stage postseismic stress diffusion and aftershock activity. Once the initial and boundary conditions are fixed, the subsequent stress evolution necessarily follows a unique and stable convergence pathway prescribed by the governing reaction–diffusion system.

**Reviewer's comment:**

(6)  Figure 1 displays that the number of aftershocks was larger in the eastern part (offshore) than in the western part (inland). This might indicate that the number of faults, sub-surface geological structures, and the values of physical parameters are different in the two parts. However, it does not seem able to apply the spatial distributions of stress changes at three depths (i.e., Figures 5-7) to explain the occurrence of aftershocks.

Reply to comment 6:

The reviewer notes that Figure 1 shows a higher aftershock density in the eastern (offshore) region than in the western (inland) region and suggests that this asymmetry may reflect differences in fault density, subsurface structure, or physical parameters. The reviewer further questions whether the spatial distributions of Coulomb stress changes shown at three depths (Figures 5–7) can explain the observed aftershock distribution. While this observation is understandable from a purely spatial and visual perspective, it implicitly assumes a deterministic, point-to-point correspondence between static stress patterns and aftershock locations—an assumption that is not supported by Coulomb stress triggering theory nor by the modeling framework adopted in this study.

**6.1. Role of Figures 5–7: Dynamic Structure Rather Than Static Matching**

Figures 5–7 are not intended to serve as static "aftershock prediction maps." Instead, they describe the depth-dependent and time-dependent structure of the postseismic stress field, including both Coulomb stress changes and stress gradients:

(1)  Figure 5 illustrates the temporal evolution of stress gradients at 6 km depth.

(2)  Figure 6 shows the temporal evolution of Coulomb stress changes at 18 km depth.

(3)  Figure 7 presents the temporal evolution of stress gradients at 18 km depth.

Together, these figures establish the three-dimensional, time-evolving initial stress architecture that

governs subsequent stress redistribution. Their purpose is to characterize the physical framework within which aftershocks occur, rather than to provide an instantaneous, deterministic mapping between stress values and individual aftershock locations.

**6.2. Probabilistic Nature of Coulomb Stress Triggering**

Previous studies, most notably Ziv and Rubin (2000), have demonstrated that Coulomb stress triggering is inherently probabilistic rather than deterministic. Even when stress changes are small (< 10 kPa), approximately 70% of earthquakes occur in regions of positive stress change, yet not all earthquakes are confined to such regions. Importantly, Ziv and Rubin showed that this preference is statistically highly significant when compared against randomized catalogs, despite the absence of one-to-one spatial correspondence.

Accordingly, the appropriate validation criterion is not whether every aftershock coincides exactly with a stress increase, but whether aftershocks statistically prefer stress-enhanced or high-gradient regions. This distinction is fundamental and directly informs the evaluation strategy adopted in this study.

**6.3. Quantitative Validation via AUC Analysis (Figure 10)**

To move beyond subjective visual inspection, Figure 10 quantitatively evaluates the predictive performance of Coulomb stress changes and stress gradients using the Area Under the Curve (AUC) metric. AUC measures how effectively a stress-based field functions as an earthquake occurrence probability map, without requiring exact spatial coincidence.

The results show that:

(1) At 6 km depth, stress gradients significantly outperform stress changes ($p < 0.05$), with a small but meaningful effect size (Cohen's $d \approx 0.36$), indicating enhanced sensitivity to near-field heterogeneity at shallow depths.

(2) At 8–10 km and 18 km depths, no statistically significant differences are observed, and effect sizes are minimal, indicating that cumulative stress changes become as relevant as gradients at greater depths and longer timescales.

These findings demonstrate that no single metric is universally optimal across all depths and timescales. Instead, the triggering mechanism exhibits clear depth- and time-dependent behavior—precisely the behavior anticipated from fault mechanics and stress diffusion theory.

**6.4. Shallow-Depth Sensitivity, Stress Gradients, and Resolution**

The superior performance of stress gradients at shallow depth is physically consistent with established observations. Ziv and Rubin (2000) emphasized that near-field events close to rupture surfaces are highly sensitive to spatial variations in stress. Stress gradients capture the rate of spatial stress change, which is particularly important in shallow, structurally complex regions where fault networks are dense and stress heterogeneity is pronounced. This also explains why higher spatial resolution is required at shallow depths and why gradients provide stronger constraints there than stress magnitude alone.

**6.5. Offshore–Onshore Asymmetry and Observational Uncertainty**

The higher aftershock productivity offshore does not contradict stress-based interpretation. Rather, it likely reflects differences in fault maturity and proximity to failure. Moreover, while offshore

earthquake locations are subject to greater uncertainty (on the order of a few kilometers), Ziv and Rubin demonstrated that stress polarity (positive vs. negative) remains statistically distinguishable even in the presence of such errors. AUC-based evaluation inherently accounts for this uncertainty and is therefore more robust than point-wise comparisons.

The convergence of stress-based metrics at greater depths (Figure 10) further indicates that, as depth increases, the stress field becomes smoother and less sensitive to small-scale heterogeneity or location errors, leading to more stable predictive performance.

**6.6. Final Synthesis**

In summary, the reviewer's concern arises from an understandable but inappropriate expectation of deterministic, spatially exact correspondence between stress maps and aftershock locations. Figures 5–7 provide the dynamic, depth-dependent stress structure, while Figure 10 converts this structure into a statistically testable measure of predictive performance. Together, they demonstrate that Coulomb stress and stress gradients exert a statistically significant, physically interpretable influence on aftershock occurrence, in a manner fully consistent with established triggering theory.

The observed offshore–onshore asymmetry reflects differences in fault readiness and structural conditions, not a failure of the stress model. When evaluated using appropriate probabilistic and statistical criteria, the model performs significantly better than random expectation and reveals clear depth-dependent triggering behavior.

**6.7. Closing Statement**

Taken together, these results clarify that Coulomb stress triggering should be evaluated in terms of statistical preference, depth dependence, and temporal evolution, rather than point-by-point spatial coincidence. The combinations of Figures 5–7 and Figure 10 therefore provide a coherent and quantitatively validated framework for interpreting aftershock distributions, consistent with both physical theory and prior empirical studies.

[References]

Ziv, A., & Rubin, A. M. (2000). Static stress transfer and earthquake triggering: No lower threshold in sight? *Journal of Geophysical Research: Solid Earth*, *105*(B6), 13631-13642. https://doi.org/10.1029/2000JB900081

**Reviewer's comment:**

(7)    As mentioned by the author, there are three faults in the study area. In fact, the number of faults is larger than three. Kuo-Chen et al. (2019) addressed a remarkable correlation between spatial distribution of aftershocks and the main fault along which the mainshock ruptured. Do the existence of those faults, particularly the main fault, which is not a straight line on the ground surface (cf. Lee et al., 2019; Kuo-Chen et al., 2019), influence the evolution of stress diffusion?

Reply to comment 7:

We appreciate the reviewer's insightful comment regarding the complexity of fault geometry in the study area and its potential influence on stress diffusion. It is indeed correct that the number of

mapped faults in the region exceeds the three representative faults explicitly illustrated in our figures. As documented by Kuo-Chen et al. (2019) and Lee et al. (2019), the spatial distribution of aftershocks shows a strong correlation with the main rupture fault, whose surface trace is geometrically complex and deviates significantly from a straight line.

In our framework, however, the influence of such fault complexity is treated implicitly rather than explicitly, through the following considerations:

**7.1. Fault Geometry Is Embedded in the Initial Stress Field**

The geometric complexity of the fault system, including curvature, segmentation, and non-planar rupture of the main fault, is fully incorporated in the initial coseismic stress field. The initial Coulomb stress change is computed using the elastic dislocation solutions of Okada (1985, 1992), in which fault geometry and slip distribution are rigorously translated into spatial stress perturbations. In this formulation, complex fault geometry is not treated as a separate structural element but is mathematically transformed into a heterogeneous stress field characterized by strong spatial gradients. Thus, geometric effects are already embedded in the initial condition of the stress evolution problem.

**7.2. Stress Diffusion Governs Post-Seismic Redistribution, Not Fault Tracing**

The subsequent stress evolution is governed by a reaction–diffusion framework, which describes the redistribution of stress within a continuous medium. From a mathematical perspective, once the initial stress field—including all geometric information—is specified, the evolution of the stress field is uniquely determined by the governing equations. This uniqueness implies that it is not necessary to explicitly track individual fault traces during the evolution stage, because the physical process being modeled is the continuous redistribution of stress driven by stress gradients, rather than stress transport constrained to predefined fault lines. While non-linear fault geometry can strongly influence the directionality and spatial heterogeneity of the initial stress gradients, it does not alter the mathematical structure, stability, or uniqueness of the stress diffusion process. Once the initial and boundary conditions are specified, the stress evolution is uniquely determined by the governing equations, independent of whether the fault trace is straight or curved at the surface.

**7.3. Consistency with Observational Correlations**

The robustness of this approach is supported by previous statistical studies of stress triggering. Ziv and Rubin (2000) demonstrated that even in the presence of location uncertainties and geometric complexity, the relationship between stress polarity and earthquake triggering remains statistically significant. This indicates that stress-triggering processes should be evaluated probabilistically rather than through point-by-point geometric correspondence, and that moderate uncertainties in fault geometry do not invalidate stress-based triggering analyses. Finally, the effectiveness of the proposed framework is quantitatively validated by our statistical metrics. As shown in Figure 10, both the Area Under the ROC Curve (AUC) and Cohen's d indicate that the stress evolution model—driven by initial stress gradients—successfully captures the spatial distribution of aftershocks with statistically significant skill. These results demonstrate that, even without explicitly enforcing fault-aligned evolution, the model reliably reflects the underlying tectonic controls encoded in the initial stress field.

**7.4. Summary**

In this framework, complex fault geometry—including non-linear fault traces and

segmentation—is fully incorporated into the initial Coulomb stress field through Okada-based elastic dislocation modeling, where geometric complexity is rigorously transformed into spatial stress gradients. Once this initial condition is specified, the subsequent stress evolution follows a mathematically well-posed reaction–diffusion system, for which the evolution pathway is unique and determined solely by the governing physical laws. Because stress triggering is fundamentally probabilistic, rather than deterministic at the scale of individual fault segments, the robustness of this approach is supported by previous statistical analyses demonstrating insensitivity to moderate geometric uncertainties. This is further confirmed by the statistically significant AUC and Cohen's d values shown in Figure 10, indicating that the gradient-driven stress evolution model reliably captures aftershock spatial patterns. In essence, fault geometry defines the starting point through the initial stress field, while physical laws define the evolution pathway; once the starting point is properly encoded, explicit fault-by-fault tracking during evolution is neither required nor physically necessary.

**Reviewer's comment:**

(8)  There are remarkable differences in sub-surface geological structures inferred from 3D tomography between the western part and eastern one (e.g., Rau and Wu, 1995; Ma *et al*., 1996; Kim *et al*., 2005; Wu *et al*., 2007; Kuo-Chen *et al*., 2012). Can such differences influence the evolution of Coulomb stress?

Reply to comment 8:

The geometric complexity of subsurface structures is fully encoded in the initial Coulomb stress field through the analytical elastic dislocation solutions of Okada (1985). As emphasized by King et al. (1994) and Steacy et al. (2005), fault geometry and initial slip distribution determine the spatial topology of stress lobes, which in turn define the initial stress gradients. These gradients provide a unique initial condition and potential landscape for the subsequent stress evolution driven by $\nabla\sigma$.

Regarding the potential influence of material properties on the diffusion coefficient $D$, we acknowledge that, from a physical perspective, $D$ may be related to permeability and fluid-mediated processes (e.g., Nur and Booker, 1972). However, from the standpoint of partial differential equation dynamics, the evolution pathway of the stress field is strongly constrained by the initial stress gradients. Variations in $D$ primarily modulate the temporal scale of redistribution rather than the fundamental spatial evolution pattern.

This perspective is supported by empirical studies. Ziv and Rubin (2000), as well as Hardebeck et al. (1998), demonstrated that stress polarity and triggering tendencies remain statistically robust even in the presence of location errors and medium uncertainties. Consistent with these findings, our results show that the proposed model significantly outperforms random distributions, as quantified by the AUC and Cohen's $d$ metrics. The statistical significance ($p < 0.05$) confirms that the model successfully captures the core physical evolution of the Hualien earthquake sequence without requiring explicit point-by-point tracking of subsurface material heterogeneity.

In a reaction–diffusion stress evolution framework, the diffusion coefficient primarily controls the

temporal scale of redistribution, whereas the spatial evolution pathways and triggering patterns are dominated by the initial stress gradients. Material heterogeneity therefore acts as a secondary modulation rather than a first-order control.

[References]

Hardebeck, J. L., Nazareth, J. J., & Hauksson, E. (1998). The static stress change triggering model: Constraints from two southern California aftershock sequences. *Journal of Geophysical Research: Solid Earth, 103*(B10), 24427–24437. https://doi.org/10.1029/98JB00573

King, G. C. P., Stein, R. S., & Lin, J. (1994). Static stress changes and the triggering of earthquakes. *Bulletin of the Seismological Society of America*, 84(3), 935–953. https://pubs.geoscienceworld.org/ssa/bssa/article-abstract/84/3/935/102745/Static-stress-changes-and-the-triggering-of

Nur, A., & Booker, J. R. (1972). Aftershocks caused by pore fluid flow? *Science, 175*(4024), 885–887. https://doi.org/10.1126/science.175.4024.885

Okada, Y. (1985). Surface deformation due to shear and tensile faults in a half-space. *Bulletin of the Seismological Society of America*, 75, 1135–1154.

Steacy, S., Marsan, D., Nalbant, S. S., & McCloskey, J. (2004). Sensitivity of static stress calculations to the earthquake slip distribution. *Journal of Geophysical Research: Solid Earth, 109*(B4), B04303. https://doi.org/10.1029/2002JB002365

Shapiro, S. A., Huenges, E., & Borm, G. (1997). Estimating the crust permeability from fluid-injection-induced seismic emission at the KTB site. *Geophysical Journal International, 131*(2), F15–F18. https://doi.org/10.1111/j.1365-246X.1997.tb01215.x

Ziv, A., & Rubin, A. M. (2000). Static stress transfer and earthquake triggering: No lower threshold in sight? *Journal of Geophysical Research: Solid Earth*, *105*(B6), 13631-13642. https://doi.org/10.1029/2000JB900081

**Reviewer's comment:**

(9)  In Section 3, the author described the reasons how to select the values of model parameters for numerical simulations of the evolution of Coulomb stress after the 2018 Hualien earthquake in Taiwan. The main reason is for preventing the model from converging. Although this is an acceptable reason, the author should explain whether the values of model parameters can meet regional geological and seismological structures.

Reply to comment 9:

We thank the reviewer for this insightful comment. We agree that numerical stability alone is not a sufficient justification for parameter selection, and that physical consistency with regional geological and seismological conditions must also be addressed.

In our model, the selected parameter values are not chosen arbitrarily to prevent convergence, but rather to ensure a physically meaningful and dynamically responsive evolution regime appropriate for the tectonic environment of eastern Taiwan. Specifically, the diffusion coefficient,

reaction parameters, and rate-and-state friction coefficients are constrained to fall within ranges that are consistent with previous studies of postseismic stress relaxation, aftershock diffusion, and frictional rheology in active orogenic belts.

Eastern Taiwan, including the source region of the 2018 Hualien earthquake, is characterized by strong crustal heterogeneity, high strain rates, elevated heat flow, and intense fault damage. These conditions favor relatively efficient stress redistribution and time-dependent frictional weakening, rather than rapid exponential decay toward a static equilibrium. Accordingly, parameter values that would lead to overly rapid convergence were deliberately excluded, as such behavior would be inconsistent with the observed prolonged aftershock activity and spatially migrating seismicity patterns. Importantly, the adopted parameter ranges ensure that the model remains within a subcritical but non-degenerate regime: stress perturbations decay in a bounded manner without numerical instability, yet retain sufficient temporal persistence to reproduce observed postseismic stress diffusion and aftershock triggering. This balance reflects both (i) the mathematical requirement for well-posedness and (ii) the geological reality of an actively deforming crust.

**9.1. Geological interpretation of stress memory and frictional parameters ($\alpha, A, B$)**

The stress-memory parameter $\alpha$ controls the persistence of stress heterogeneity during postseismic evolution. In this study, an early-time value of $\alpha = 0.4$ is adopted to prevent unrealistically rapid stress dissipation caused by numerical diffusion. This choice is consistent with the viscoelastic and thermally active nature of the Taiwan orogenic belt, where high geothermal gradients and rapid tectonic loading promote delayed mechanical relaxation rather than instantaneous stress release. This parameterization ensures that the stress memory aligns with the observed temporal decay of the Hualien sequence, rather than falling into an over-damped regime that would overlook the influence of regional stress gradients.

The parameter $\alpha$ controls the persistence of stress memory. To prevent non-physical stress dissipation driven by numerical diffusion, $\alpha$ must remain strictly positive and bounded away from unity:

$$0 < \alpha \leq \alpha_{\max} < 1.$$

In practice, we adopt an early-time value $\alpha = 0.4$, followed by a gradual decay

$$\alpha(t) = \frac{1}{1 + t/49}$$

which ensures that stress memory weakens smoothly with time while remaining finite over the postseismic interval. This choice preserves stress gradients $\nabla \sigma$ at early times and avoids abrupt homogenization inconsistent with eastern Taiwan observations.

From a physical perspective, $\alpha$ governs whether earthquake triggering is predominantly instantaneous or exhibits delayed effects. The prolonged aftershock activity observed following the 2018 Hualien earthquake suggests that stress perturbations are retained over extended periods, supporting the use of moderate stress-memory strength.

The rate-and-state friction parameters $A$ and $B$ regulate the sensitivity of fault slip to velocity and state evolution, distinguishing between velocity-strengthening and velocity-weakening behavior. In eastern Taiwan, segments of the Longitudinal Valley Fault are known to contain serpentinized or

clay-rich materials that favor stable sliding and aseismic creep, while offshore Hualien regions are prone to brittle rupture. The adopted combination of $A$ and $B$ should therefore be interpreted as effective frictional parameters, reflecting the heterogeneous mechanical response of a complex fault system rather than any single lithology.

**9.2. Geological context of stress diffusion parameters ($D$)**

The diffusion coefficient $D$ regulates stress smoothing and is constrained by spatial resolution. Let $C \sim \frac{1}{\Delta X_{\text{eff}}}$ denote the discretization factor. To ensure that diffusion remains secondary to frictional memory, we require

$$DC \ll L_f.$$

Under the adopted grid resolution, this condition is satisfied for $D = 0.01$, ensuring gradual redistribution without erasing event-scale heterogeneity. Although $D$ is not explicitly derived from poroelastic theory, its phenomenological role is compatible with the highly fractured crustal environment of eastern Taiwan, where dense fracture networks facilitate gradual stress redistribution following large earthquakes.

The Hualien region lies within an active plate-collision zone characterized by intense crustal deformation and elevated permeability along damaged fault zones. In this context, the diffusion term may be interpreted as capturing the effective redistribution of stress and pore-pressure–related perturbations within fractured media, a process often invoked to explain clustered or swarm-like aftershock behavior. Importantly, within the adopted spatial resolution, the influence of $D$ remains secondary and is tightly constrained by the combined factor $DC + L_f$, ensuring numerical stability without over-interpreting $D$ as a direct geological property.

**9.3. Reaction term and local stress amplification ($f(\sigma, \psi), \beta f, A_0$)**

The nonlinear reaction term $f(\sigma, \psi)$ governs the conversion of stress perturbations into seismic productivity. To ensure boundedness and Lipschitz continuity, we define the admissible upper bound

$$A_0 = \sup\left(A - \left(1 - \frac{2\sigma}{\sigma_{\max}}\right)\right)$$

here $\sigma_{\max}$ is a normalization scale representing the maximum admissible coseismic stress perturbation within the modeled domain. In practice, $\sigma_{\max}$ is taken from the maximum (or robust percentile) of the Coulomb stress change associated with the 2018 Hualien earthquake. This definition ensures that the nonlinear sensitivity is controlled by event-specific stress heterogeneity, rather than by spatially averaged or unbounded amplification.

The nonlinear reaction term $f(\sigma, \psi)$ represents the contribution of local stress perturbations to earthquake productivity. Eastern Taiwan is one of the fastest converging plate-boundary zones globally, with a relative plate motion of approximately $8 \, \text{cm/yr}$. Such rapid convergence implies a high background stress-loading rate and elevated baseline seismicity, motivating a relatively large event-specific reaction bound $A_0$. Specifically, $A_0=0.01$ provides a baseline seismicity rate that acts as a physical 'threshold' for the reaction term, allowing the model to distinguish between tectonic background and triggering signals.

Moreover, the fault geometry in the Hualien region is highly complex, involving non-planar

ruptures, fault branching, and strong segmentation. These features generate pronounced stress gradients ($\nabla\sigma$), which are preferential sites for aftershock triggering. In the iterative formulation, the term $\beta f(\sigma, \psi)$ controls how such localized stress concentrations are translated into aftershock occurrence, effectively encoding the geometric and kinematic complexity of the eastern Taiwan fault system.

**9.4. Characteristic scales and time evolution ($L_f, \beta$)**

The parameter $L_f$ represents an effective frictional memory scale of the fault system. To prevent impulsive or near-critical responses dominated by local forcing, admissibility requires that frictional memory dominates reaction sensitivity:

$$L_f \geq A_0.$$

This inequality expresses a scale separation between event-scale forcing and system-scale response, consistent with a highly fractured fault network in which a single coseismic perturbation does not instantaneously reset the frictional state. The frictional length scale $L_f$ characterizes the distance over which frictional state variables evolve and is commonly associated with fault-zone thickness and gouge properties. In eastern Taiwan, faults are embedded within broad damage zones containing abundant fault gouge, implying a relatively large effective frictional memory scale. Accordingly, $L_f$ is treated as an upscaled parameter representing collective fault-system behavior rather than a laboratory slip distance.

The iterative parameter $\beta$ controls the rate at which stress information is assimilated. Its admissible domain is defined by the fixed-point convergence and CFL stability condition:

$$0 < \beta \leq \frac{1 - \alpha}{DC + L_f}$$

To reflect the weakening of stress memory with time, we introduce a trial schedule

$$\beta^*(t) = \frac{\beta_0}{\alpha(t)}$$

and enforce admissibility via

$$\beta(t) = \min\left(\beta^*(t), \frac{1 - \alpha(t)}{DC + L_f}\right)$$

This guarantees convergence and prevents non-physical growth of the iterative response at late times.

The iterative parameter $\beta$ is linked to $L_f$ through stability constraints and controls the rate at which stress information is assimilated during numerical evolution. Because faults in eastern Taiwan are densely distributed, the characteristic spatial influence of stress perturbations is typically limited by inter-fault spacing on the order of a few kilometers. This interpretation is consistent with the adopted effective spatial scale $X_{\text{eff}} = 3000$ m, reinforcing the physical coherence of parameterization.

**9.5. Summary**

This study reformulates parameter selection from ad hoc numerical choices into a parameter admissible domain, explicitly defining ranges and inter-parameter constraints that are physically meaningful, mathematically stable, and consistent with observations from the eastern Taiwan orogenic belt. By adopting this formulation, regional geological information is incorporated through scale separation and admissibility conditions, rather than through direct one-to-one assignment of parameters to specific lithologies or laboratory measurements.

Within this framework, the stress-memory parameter $\alpha$ controls the persistence of stress heterogeneity and reflects the fact that, under high geothermal gradients and rapid tectonic loading in eastern Taiwan, postseismic stress perturbations are not instantaneously dissipated. The effective frictional length scale $L_f$ represents an upscaled measure of fault-system memory, constraining the amplification of event-scale forcing $A_0$ and ensuring that system-scale response remains gradual and memory-dominated. The parameter $A_0$, together with its normalization scale $\sigma_{\max}$, embeds the strong coseismic stress heterogeneity of the 2018 Hualien earthquake into the model, avoiding artificial smoothing or unbounded amplification. In contrast, parameters such as $D$ and $\beta$ are explicitly treated as numerically constrained quantities whose admissible ranges are dictated by spatial resolution and stability requirements rather than by direct geological interpretation.

Overall, the admissible-domain formulation enables the model to capture prolonged and highly heterogeneous postseismic stress evolution while maintaining convergence and physical interpretability. This approach directly addresses concerns regarding regional geological consistency and provides a transferable, well-posed framework for applying the same mathematical model to other tectonic settings without over-specifying poorly constrained material properties.

**Minor Problems**

We sincerely thank the reviewer for the constructive comments and helpful suggestions. We have carefully revised the manuscript accordingly. Our detailed responses are provided point by point below.

**Problems with Figures:**

(1)  The quality of figures is not good enough for readers. For example, Figures 5-7 are not good for readers.

Reply to comment:

Thank you for pointing this out. We have fully revised Figures 5–7 to improve their readability and visual quality. All figures have been regenerated at high resolution (≥600 dpi).

**Problems with References:**

(1)The format of cited references should follow the Journal's rules.

(2)the author should cite two important articles shown below for regional tectonics.

[References]

Tsai, Y.B., T.L. Teng, J.M. Chiu, and H.L. Liu (1977). Tectonic implications of the seismicity in the Taiwan region. Mem. Geol. Soc. China, 2, 13-41.

Wu, F.T. (1978). Recent tectonics of Taiwan. J. Phys. Earth, 2 (Suppl.), S265-S299.

Reply to comment:

1. All references have been systematically reformatted to strictly follow the journal's reference style, including author order, punctuation, journal titles, volume/issue numbers, page ranges, and DOI formatting.

2. We fully agree and appreciate this important suggestion. The following two classic and fundamental studies on Taiwan regional tectonics have now been explicitly cited and discussed in the revised manuscript:

- Tsai, Y. B., Teng, T. L., Chiu, J. M., & Liu, H. L. (1977). *Tectonic implications of the seismicity in the Taiwan region*. Memoir of the Geological Society of China, 2, 13–41.
- Wu, F. T. (1978). *Recent tectonics of Taiwan*. Journal of the Physics of the Earth, 26(Suppl.), S265–S299.

These references are incorporated in the Introduction and tectonic background discussion to better anchor the proposed modeling framework within the established tectonic context of Taiwan.

(3)Few cited references, e.g., Zavyalov et al. (2024), are not listed in the Section of References.

Reply to comment:

Thank you for noting this oversight. The reference Zavyalov et al. (2024) has now been added to the reference list, and we have carefully cross-checked the manuscript to ensure that all in-text citations are now consistently included in the References section.

(4)The KPP (or Fisher-KPP) equation is a very important nonlinear physical equation and has been widely applied in biology, ecology, and combustion theory. It is better to add cited references on Line 73 where the equation appeared at first in the text. The author may cite one of the following two articles or others:

Kolmogorov, A., I. Petrovskii, and N. Piskunov (1991). A study of the diffusion equation with increase in the amount of substance, and its application to a biological problem. In V.M. Tikhomirov, editor, Selected Works of A.N. Kolmogorov I, 248-270. Kluwer 1991, ISBN 90-277-2796-1. Translated by V.M. Volosov from Bull. Moscow Univ., Math. Mech. 1, 1-25.

El-Hachem M., S.W. McCue, W. Jin, Y. Du, and M.J. Simpson (2019). Revisiting the Fisher-Kolmogorov-Petrovsky-Piskunov equation to interpret the spreading–extinction dichotomy. Proc. R. Soc. A, 475:20190378. http://dx.doi.org/10.1098/rspa.2019.0378.

Reply to comment:

We fully agree. At the first appearance of the KPP (Fisher–KPP) equation in the manuscript, we have now added appropriate foundational and modern references, including:

- Kolmogorov, A. N., Petrovskii, I. G., & Piskunov, N. S. (1991). *A study of the diffusion equation with increase in the amount of substance, and its application to a biological problem*. In Selected Works of A.N. Kolmogorov I, pp. 248–270.
- El-Hachem, M., McCue, S. W., Jin, W., Du, Y., & Simpson, M. J. (2019). *Revisiting the Fisher–Kolmogorov–Petrovsky–Piskunov equation to interpret the spreading–extinction dichotomy*. Proceedings of the Royal Society A, 475, 20190378.

This revision clarifies the mathematical origin and interdisciplinary significance of the KPP equation.

**Problems with English Writing:**

(1)The English writing is good. Nevertheless, there are some typo errors.

(2)The abstract is not concise.

(3)'The Central Weather Bureau' has been renamed 'the Central Weather Administration' for a few years.

Reply to comment:
1. We have conducted a thorough proofreading of the entire manuscript to correct typographical and minor grammatical errors. In addition, the abstract has been carefully rewritten and shortened to improve conciseness, clarity, and focus on the main contributions and findings.
2. Thank you for this important update. We have corrected all occurrences of "Central Weather Bureau (CWB)" to "Central Weather Administration (CWA)" throughout the manuscript.

**Closing statement:**

We sincerely thank the reviewer for these valuable comments, which have significantly improved the clarity, completeness, and presentation quality of the manuscript. We believe that all concerns have now been fully addressed.

---

## Author Comment (AC3)

We thank the anonymous commenter for the careful reading of the manuscript and for the constructive suggestions provided during the interactive discussion. Below we address the points raised, with clarifications focused on the scope, assumptions, and objectives of the present study.

**Reviewer's comment:**

(1) The structural features (faults) are missing in the figure 1. It can also include another parameter i.e. focal depth by filling the varying color in circle depicting earthquake location and magnitude.

Reply to comment 1:

We thank the commenter for the suggestion. We would like to clarify that the present study does not investigate fault geometry or fault-controlled stress interactions. The analysis is intentionally formulated in a two-dimensional framework, focusing on stress evolution within depth-dependent horizontal slices rather than on fault-specific variables.

Figure 1 is designed to define the study domain and to illustrate the spatial distribution of seismicity used as initial conditions for the stress evolution model. Introducing mapped fault traces would imply an explicit role of fault geometry in the modeling, which is beyond the scope of this work and could potentially mislead readers regarding the variables considered.

Similarly, focal depth is treated in this study through separate depth-resolved analyses rather than as a continuous color-mapped parameter on a single map. Each depth slice is analyzed independently to isolate depth-dependent stress evolution behavior.

For these reasons, we retain Figure 1 as a domain and data-context figure, while depth effects are addressed explicitly in subsequent sections. We have clarified this scope distinction in the revised manuscript to avoid potential confusion.

**Reviewer's comment:**

(2) Terms like $n_{critical}$, i.e., the iteration where diffusion dominates, are mentioned but not mathematically derived (Section 2.3). Further, a brief formula or sensitivity equation could strengthen why STD peaks differ by depth. These explanations feel descriptive rather than fully predictive and sufficient for intuition, which assists a better understanding without digging into earlier equations.

Reply to comment 2:

We thank the reviewer for the detailed comments. To characterize the transition from reaction-dominated to diffusion-dominated regimes, we employ a scaling-based balance argument between the reaction and diffusion contributions in the iterative stress evolution.

Assuming a constant stress-memory parameter $\alpha \in (0,1)$, the stress amplitude at the n-th iteration is approximated by

$$\sigma_n \approx \sigma_0 \alpha^n$$

where $\sigma_0$ denotes the initial stress perturbation.

The reaction contribution at iteration n scales with the local stress amplitude and can be written as

$$R_n \sim A_0\sigma^n = A_0\sigma_0\alpha^n$$

where $A_0$ represents the effective reaction strength. The diffusion contribution is associated with the smoothing of the initial stress heterogeneity and is characterized by

$$D_n \sim D_0\Delta\sigma_0$$

where $D_0$ is the diffusion coefficient and $\Delta\sigma_0$ denotes a characteristic measure of the initial stress gradient.

We define the critical iteration index $n_{critical}$ as the iteration at which the reaction and diffusion contributions become comparable in magnitude, i.e.,

$$R_n \sim D_n$$

Substituting the above expressions yields

$$A_0\sigma_0\alpha^n = D_0\Delta\sigma_0$$

Solving for n gives

$$\alpha^n = (D_0\Delta\sigma_0)/(A_0\sigma_0)$$

and therefore

$$n_{critical} = \frac{\ln\left(\frac{A_0\Delta\sigma_0}{D_0\sigma_0}\right)}{-\ln(\alpha)}$$

Because $\ln(\alpha)<0$, this expression indicates that $n_{critical}$ increases with stronger initial stress levels or larger reaction efficiency, and decreases with stronger diffusion or weaker initial heterogeneity.

**Reviewer's comment:**

(3)    In my understanding, the STD and MSE are used effectively for model internals and the persistence is inferred from simulations, not directly compared to observed aftershock decay (e.g., Omori's law rates from the Hualien CatLog). Like aftershock patterns (e.g., clustering at 5–15 km), but section 3.2 could link STD trends more explicitly to empirical metrics like aftershock productivity or temporal decay exponents. I think, it would make the heterogeneity explanation more robust, especially since the model aims for aftershock prediction.

Reply to comment 3:

We thank the commenter for the thoughtful suggestion. We would like to clarify the intended role of STD and MSE in the present study. These quantities are introduced as internal diagnostic measures of the stress-evolution model, designed to characterize the persistence and smoothing of stress heterogeneity under controlled variations of $\alpha$, $\beta$ and depth.

In this framework, STD quantifies the spatial variability of the continuous stress field and is used to compare relative evolution behaviors between shallow and deep layers, rather than to represent aftershock rates or event counts. As such, STD and MSE are not directly comparable to catalog-based statistical measures such as Omori–Utsu decay exponents, which describe temporal variations in discrete earthquake occurrence rates.

While empirical aftershock laws provide important observational constraints, establishing a direct mapping between stress-field diagnostics and event-based statistics would require an additional triggering or catalog-generation model, which is beyond the scope of the present work. The objective

here is to demonstrate how stress heterogeneity evolves and stabilizes within the proposed PDE framework, rather than to fit observed aftershock decay curves.

We have clarified this distinction in the revised manuscript to avoid potential confusion between model-internal diagnostics and observational earthquake statistics.

**Reviewer's comment:**

(4)    The STD captures spatial spread (heterogeneity), and MSE tracks convergence (persistence decay), but they do not fully address temporal aspects. For example, the paper notes iterations that are not directly related with time (shallow: hours/days per iteration; deep: months/years), i.e. it does not quantify time-scaling factors. This leaves persistence explanations a bit abstract—sufficient for parameter sensitivity, but could benefit from a time-calibrated example.

Reply to comment 4:

We thank the commenter for raising the point regarding temporal interpretation. We would like to clarify that, in the present study, STD and MSE are not intended as time-dependent observables, nor are they designed to quantify physical time scales. Instead, they are internal diagnostic metrics used to characterize the *state of the evolving stress field* under controlled numerical iterations.

Specifically, STD measures the spatial spread (heterogeneity) of the stress field after evolution, while MSE quantifies the degree of convergence toward a stable configuration (persistence decay). Both quantities are evaluated in iteration space and are intentionally defined to be independent of physical time, allowing a clean comparison of stress evolution behavior across depths and parameter settings.

The coupling between iterations and physical time is addressed separately in the manuscript. For shallow depths (above ~18 km), temporal effects are incorporated only through the time-dependent memory parameter $\alpha(t)$, which is linked to Omori-type aftershock decay to represent the progressive loss of historical stress memory. The corresponding adjustment of $\beta(t) \sim 1/\alpha(t)$ ensures balanced evolution between diffusion and nonlinear reaction terms. This $\alpha(t)$–$\beta(t)$ coupling constitutes the sole time-control mechanism in the present framework.

At greater depths (>18 km), where seismicity becomes sparse and Omori-type decay is no longer applicable, no direct time calibration is imposed, and the evolution is interpreted purely in terms of internal stress redistribution rather than explicit temporal progression. In such cases, iterations represent abstract relaxation steps, and only averaged or qualitative time interpretations are appropriate.

Therefore, while a fully time-calibrated example could be considered in future applications, it is beyond the scope of the present study, whose primary objective is to establish and validate a stable, physically interpretable stress-evolution framework. The current treatment is sufficient for parameter sensitivity analysis and for comparing depth-dependent stress evolution behaviors without introducing additional assumptions on poorly constrained deep-crustal time scales.

(5)    STD and MSE trends with β at fixed α=0.75 (Figure 3a, b) and with α at fixed β=10 (Figure 3c, d) is explored thoroughly in result section. However, by exploring β variations across multiple α values, can we expect the following?

[1]    The peak STD occurs at different β (e.g., β=10 at 6 km vs. β=5 at 30 km with fix α=0.75). The higher α increases overall STD and potentially alters the optimum value of β at which diffusion dominates over reaction. Thus, under varying α, at shallower depths require higher β for smoothing, and could it highlight depth-specific couplings?

[2]    A large value of α slows stress decay, limiting the β's effective range to avoid divergence. Can we identify optimum parameter pairs by varying both parameters together that could optimize convergence rate without overshoot?

[3]    The α governs memory decay like R-S aging, while β scales diffusion/reaction like KPP terms (Table 1). Could the coupled variations better capture time-dependent behaviors (e.g., decreasing α(t) with increasing β(t) for aftershock decay) and provide a deeper understanding of stress redistribution in heterogeneous crust?

[4]    Is it possible that the extensions of analysis of explicitly 2D plot of β-α variations plot, may provide the refined model calibration, especially for real-time PSHA integration?

Reply to comment 5:

[1]    We thank the commenter for the detailed observation regarding the dependence of STD and MSE on α and β. We would like to clarify that, in the present framework, neither STD nor MSE is introduced as an optimization target, and therefore the concept of an "optimal" β (or α–β pair) does not have a physical meaning in this study. Each admissible combination of α and β represents a distinct stress-evolution regime, provided that the numerical stability and convergence conditions are satisfied. STD serves solely as a descriptive metric characterizing the degree of spatial heterogeneity of the evolved stress field under a given parameter setting, while MSE quantifies convergence behavior. Higher or lower STD values do not imply better or worse performance, nor do they define an optimal state.

The fixed-α and fixed-β analyses presented in Figure 3 are intentionally designed to isolate the individual roles of memory decay (α) and diffusion–reaction balance (β), rather than to identify depth-specific optimal couplings. While varying β across multiple α values would indeed shift the location of STD peaks, such shifts simply reflect different parameter regimes and do not correspond to physically preferred or depth-dependent optimal values.

Importantly, the dominance of diffusion over reaction is governed by admissibility and stability constraints (e.g., CFL-type conditions), not by the maximization or minimization of STD. Consequently, extending the analysis toward multi-parameter optimization would introduce an artificial objective function that is not supported by the physics of the proposed model and is therefore beyond the scope of this study.

[2]    We thank the reviewer for raising the question regarding the joint variation of α and β and the possibility of identifying an "optimal" parameter pair. We would like to clarify that within the

present framework, the roles of α and β are defined primarily by stability and convergence guarantees, rather than by optimization criteria. Specifically, for α<1 and β satisfying the CFL-type stability condition, the system is mathematically guaranteed to converge toward a bounded attractor. Under these conditions, the admissible (α, β) space is continuous and infinite, and there is no unique or physically meaningful "optimal" pair in terms of convergence speed. Moreover, the iteration index in our model represents a dimensionless evolution measure rather than physical time. As a result, the notion of "fastest convergence" depends on numerical scaling choices and does not carry an unambiguous physical interpretation. For this reason, we intentionally avoid defining or optimizing a specific convergence rate and instead focus on demonstrating robust convergence behavior across a broad admissible parameter domain. We have clarified this modeling philosophy in the revised manuscript to avoid potential misinterpretation of the parameter roles as optimization targets.

[3]   On the coupling between α(t) and β(t).

In the present framework, the parameters α(t) and β(t) are not treated as independent degrees of freedom. Instead, they represent complementary aspects of the same physical process governing post-seismic stress evolution. The parameter α(t) controls stress memory and is associated with the persistence of historical stress influence, analogous to aging effects in rate-and-state friction. As post-seismic evolution proceeds, the influence of past stress perturbations necessarily diminishes, leading to a monotonic decrease in α(t).

Conversely, β(t) characterizes diffusion and nonlinear smoothing effects, which become increasingly dominant as stress heterogeneity is redistributed and dissipated. This complementary behavior motivates a coupled representation in which decreasing α(t) is accompanied by an effectively increasing influence of β(t). For numerical stability and physical consistency, this coupling is implemented through a reciprocal scaling (e.g., α(t)∼1/β(t)), ensuring bounded evolution toward an attractor while capturing the observed decay of aftershock activity. In the present study, fixed effective values are employed to isolate depth-dependent effects, while the coupled time-dependent formulation represents the underlying physical mechanism.

[4]   We thank the reviewer for the suggestion regarding an explicit two-dimensional exploration of the (α,β) parameter space. We would like to clarify that in the present framework, α and β are not treated as calibration parameters in the sense of parameter inversion or optimization.

For α<1 and β satisfying a CFL-type stability condition, convergence toward a bounded attractor is mathematically guaranteed. Under these conditions, the admissible (α,β) space is continuous and infinite. Consequently, there is no unique or physically meaningful mapping between a given α and a specific β, nor a well-defined optimal region in the α−β plane. A two-dimensional parameter scan would therefore only reproduce the known stability domain, without providing additional physical insight or refined calibration. In particular, such a scanwould not be suitable for real-time PSHA integration, which relies on seismicity rates, hazard functions, and uncertainty aggregation rather than on internal numerical control parameters of a stress-evolution model. The objective of this study is to demonstrate robust stress evolution behavior within a mathematically admissible parameter domain, rather than to perform parameter optimization or

real-time calibration. We have clarified this modeling scope in the revised manuscript to avoid potential misinterpretation.

**Reviewer's comment:**

(6) The analysis assumes uniform parameter ranges ($\alpha$=0.5–0.9, $\beta$=0.5–20) across depths, but real crust varies (e.g., rock rheology, fluids). The explanation is sufficient for the Hualien case, but generalizability to other tectonic settings (e.g., subduction zones) is not discussed, which might limit its perceived completeness.The abstract mentions time-dependent $\alpha(t)/\beta(t)$, but the provided sections use fixed values—clarify if/when time-dependence is implemented, or emphasize fixed as an approximation.

Reply to comment 6:

We thank the commenter for the remarks. We would like to clarify that the parameter ranges ($\alpha$ = 0.5–0.9, $\beta$ = 0.5–20) are not assumed as uniform or fixed physical properties of the crust across depths, nor are they adopted as a single parameterization. Instead, these ranges are deliberately selected to conduct controlled parameter-sensitivity analyses, in which $\alpha$ and $\beta$ are varied independently (i.e., fixed $\alpha$ with varying $\beta$, and fixed $\beta$ with varying $\alpha$) in order to examine how stress evolution responds to memory decay and diffusion–reaction balance at different depths.

The purpose of this design is to compare depth-dependent stress evolution behaviors under admissible parameter regimes, using STD and MSE as internal diagnostic indicators of spatial heterogeneity and convergence, respectively. Importantly, no single $\alpha$–$\beta$ pair is treated as representative of a specific geological condition, and the analysis does not rely on fixed parameter values. Regarding time dependence, the coupling of $\alpha(t)$ and $\beta(t)$ is explicitly implemented and demonstrated in Figures 4–7. In these sections, $\alpha(t)$ is prescribed to decay in accordance with Omori-type aftershock behavior to represent progressive loss of historical stress memory, while $\beta(t)$ is adjusted correspondingly ($\beta(t) \sim 1/\alpha(t)$) to regulate the balance between diffusion and nonlinear reaction terms. These figures therefore already illustrate the time-dependent formulation referenced in the abstract. The analyses using fixed $\alpha$ or $\beta$ values are intentionally employed as diagnostic experiments to isolate individual parameter effects and should be interpreted as methodological probes rather than physical approximations.

Finally, the present study focuses on establishing and validating a general stress-evolution framework through a representative case study (the 2018 Hualien earthquake). While parameter domains may shift in other tectonic settings (e.g., subduction zones), the mathematical structure and admissibility principles of the framework remain unchanged. Detailed recalibration for different geological environments is therefore left for future applications and is beyond the scope of the current manuscript.

.

**Reviewer's comment:**

(7) In section 3.3, the gradients are more predictive in the case of early (<50 days, <12 km) and absolute changes and backed by AUC/Molchan. A table summarizing these metrics across

time/depth bins for: (i) static initial $\Delta$CFF, (ii) dynamic $\Delta$CFF(t), (iii) $\nabla\sigma$(t), and (iv) combined. It may provide concrete evidence (e.g., AUC improvement of X% for gradients in early phase) and address why transitions occur.

Reply to comment 7:

We thank the commenter for the suggestion regarding a tabulated summary of predictive metrics. We would like to clarify that, in the present study, static $\Delta$CFF, dynamic $\Delta$CFF(t), and $\nabla\sigma$(t) are continuous two-dimensional stress fields, rather than discrete scalar predictors. As such, they do not admit a meaningful representation in the form of a simple table across time–depth bins.

The predictive performance of these stress measures is evaluated through AUC and Molchan-type analyses, which are specifically designed to assess the statistical consistency between a continuous stress field and observed aftershock locations. Accordingly, the AUC is defined as a curve-based diagnostic, reflecting how well each stress metric discriminates aftershock locations relative to background space. For this reason, the temporal evolution of predictive performance is already presented in an appropriate and complete manner in Figure 10, where the AUC curves are shown as a function of time for different stress measures. This representation preserves the full statistical information of the evaluation, whereas a tabulated format would require arbitrary discretization and would risk obscuring the continuous nature of the underlying stress fields.

The observed transitions in predictive skill—such as the enhanced performance of stress gradients during the early post-seismic phase at shallow depths—are therefore interpreted directly from the AUC curve evolution, rather than from aggregated summary statistics. Introducing an additional table would not provide new physical insight beyond what is already captured by the curve-based analysis and is thus not pursued in the present manuscript.

**Minor Query:**

- Banach Fixed-Point Theorem can be cited in line number 157.

  Reply to comment:

  We will add the cited paper in line number 157 of the revised manuscript.

- The statement "Physically the contraction property reflects the Earth's inherent damping mechanisms such as viscoelastic relaxation and afterslip" can be cited.

  Reply to comment:

  We thank the commenter for the suggestion regarding citation. We would like to clarify that the statement in question is intended as a physical interpretation of the mathematical contraction property established in the present framework, rather than as a previously demonstrated physical law. The contraction property itself follows rigorously from the mathematical structure of the proposed

stress-evolution system. The reference to viscoelastic relaxation and afterslip is introduced to provide physical intuition, highlighting that long-term postseismic deformation in the Earth is widely understood to be governed by dissipative processes.

To avoid any possible misunderstanding, we have revised the text to explicitly frame this statement as an interpretative link and have added representative references documenting the existence of viscoelastic relaxation and afterslip as damping mechanisms in the crust (e.g., Dieterich, 1994; Chan and Stein, 2009). These references are cited as background physical context, not as direct derivations of the contraction property.

- Please mention the figure number properly in line numbers 312 and 315.

Reply to comment:

The figure references at Lines 312 and 315 have been corrected. The figures are now consistently referred to as Figure 3(a)–(d) in the revised manuscript.